# Deep learning-based detection and segmentation of diffusion abnormalities in acute ischemic stroke

Chin-Fu Liu[1,2], Johnny Hsu[3], Xin Xu[3], Sandhya Ramachandran[1,2], Victor Wang [1,2], Michael I. Miller[1,2,4], Argye E. Hillis[5,6], Andreia V. Faria [3✉] & The STIR and VISTA Imaging investigators*

## Abstract

**Background** Accessible tools to efficiently detect and segment diffusion abnormalities in acute strokes are highly anticipated by the clinical and research communities.

**Methods** We developed a tool with deep learning networks trained and tested on a large dataset of 2,348 clinical diffusion weighted MRIs of patients with acute and sub-acute ischemic strokes, and further tested for generalization on 280 MRIs of an external dataset (STIR).

**Results** Our proposed model outperforms generic networks and DeepMedic, particularly in small lesions, with lower false positive rate, balanced precision and sensitivity, and robustness to data perturbs (e.g., artefacts, low resolution, technical heterogeneity). The agreement with human delineation rivals the inter-evaluator agreement; the automated lesion quantification of volume and contrast has virtually total agreement with human quantification.

**Conclusion** Our tool is fast, public, accessible to non-experts, with minimal computational requirements, to detect and segment lesions via a single command line. Therefore, it fulfills the conditions to perform large scale, reliable and reproducible clinical and translational research.

## Plain language summary

Determining the volume and location of lesions caused by acute ischemic strokes - in which blood flow is restricted to part of the brain - is crucial to guide treatment and patient prognosis. However, this process is time-consuming and labor-intensive for clinicians. Here, using brain imaging datasets from patients with ischemic strokes, we create an artificial intelligence-based tool to quickly and accurately determine the volume and location of stroke lesions. Our tool outperforms some similar existing approaches, it is fast, publicly available, accessible to non-experts, and it runs on normal computers with minimal computational requirements. As such, it may be useful both for clinicians treating patients and researchers studying ischemic stroke.

[1] Center for Imaging Science, Johns Hopkins University, Baltimore, MD, USA. [2] Department of Biomedical Engineering, Johns Hopkins University, Baltimore, MD, USA. [3] Department of Radiology, School of Medicine, Johns Hopkins University, Baltimore, MD, USA. [4] Kavli Neuroscience Discovery Institute, Johns Hopkins University, Baltimore, MD, USA. [5] Department of Neurology, School of Medicine, Johns Hopkins University, Baltimore, MD, USA. [6] Department of Physical Medicine & Rehabilitation, and Department of Cognitive Science, Johns Hopkins University, Baltimore, MD, USA. *A list of authors and their affiliations appears at the end of the paper. ✉email: afaria1@jhmi.edu

Stroke is a major cause of death and long-term disability in US, with increasing mortality rates among middle-aged adults[1]. Defining the stroke core in diffusion-weighted images (DWIs), a magnetic resonance imaging (MRI) sequence highly sensitive to the acute lesion, is a major benchmark for acute treatment. The importance of fast locating and objectively quantifying the acute damage has been reinforced by many stroke trials[2–5]. Clinical research also relays on objective quantification of lesions to access brain function (as lesion-symptoms studies[6]), to stratify populations in clinically relevant groups, and to model prognosis[7–10]. Due to the great variability of stroke population and lesions in both biological and technical aspects, stroke research depends on large datasets and, consequently, on automated tools to process them with accuracy and high levels of reproducibility. Therefore, accessible tools that can fast and efficiently detect and segment diffusion abnormalities in acute strokes are well anticipated by the clinical and research communities.

Traditional machine learning algorithms use texture features[11], asymmetric features[12], abnormal voxel intensity on histogram[13] or probabilistic maps of populations[14,15], or advanced statistical models such as support vector machine[16] and random forest[17], to detect and segment diffusion abnormalities in acute strokes. Although these methods represent resourceful statistical approaches and can work reasonably in high-resolution images and homogeneous protocols, the low-level features these methods utilize such as intensity, spatial, and edge information reduce their capability to capture the large variation in lesion pattern, especially in typical clinical, low-resolution, and noisy data.

As the prosperous progress in deep learning (DL) in the past two decades, several DL neural network models[18–20], such as the convolutional neural networks (CNNs), performed better brain lesion detection and segmentation than traditional methods. A landmark was the introduction of UNet[21], to re-utilize coarse semantical features via skip connections from encoders to decoders in order to predict better segmentation. Further developments of UNet variants, such as Mnet, DenseUnet, Unet++, and Unet3+[22–24] optimized the features utilization. The emergence of attention-gate techniques[25–27] conditioned networks to focus on local semantical features. Recent studies applied "attention UNets", for example, to predict final ischemic lesions from baseline MRIs[28,29]. Nevertheless, segmenting medical images with 3D networks still remains a challenge.

The 3D networks suffer from issues like gradient vanishing and lack of generalization, due to their complex architectures. Training a large number of parameters requires a large number of training samples, which is not typically available when utilizing real clinical images. As a result, most prior DL methods[20,30,31] were developed in 2D architectures, ignoring the 3D contextual information. In addition, they were mostly trained and/or tested on 2D lesioned brain slices; therefore, they are not generalizable in clinical applications where there is no prior information about the lesion location. More recently, "2.5D" networks[32], 3D dual-path local patch-wise CNNs, DeepMedic[31], and other CNNs proposed in the Ischemic Stroke Lesion Segmentation (ISLES) challenge[18], aimed to improve local segmentation. However, they still do not fully utilize the whole-brain 3D contextual information which might lead to more false positives, especially in "lesion-like" artifacts commonly seen in clinical DWIs. Furthermore, previous networks were mostly evaluated on segmenting late subacute and chronic ischemic abnormalities in high-resolution MRIs, with research-suited homogenous protocols[32–34] which do not reflect the challenges of real clinical data. To our best knowledge, Zhang et al.[35] were the first to reveal the potential of the 3D whole-brain dense networks on low-resolution clinical acute lesion segmentation. However, the relatively modest sample size utilized (242 clinical low-resolution images for training, validating and testing the models, plus 36 downsampled high-resolution images to test for generalization), compared to the large amounts of trainable parameters and high variations of stroke biological features, raises questions about the generalization and robustness of the derived models.

The requirement of large amounts of data for training and testing DL models is a common challenge for achieving good generalization and efficiency in practical applications. Artificial data augmentation does not necessarily mirror the biological heterogeneity of strokes, and imperfectly reflects the noise, low resolution, and technical variability of clinical data. Models developed in modest and homogeneous samples, or in artificial augmented data, might be less efficient in heterogeneous and noisy clinical samples[36]. Lastly, the lack of accessibility reduces the potential generalization and translation of previous methods: currently, there is no platform that allows regular users (not imaging experts) to readily and rapidly detect and segment diffusion abnormalities in acute stroke clinical images.

In this study, we developed DL network ensembles for the detection and segmentation of lesions from acute and early subacute ischemic strokes in brain MRIs. Our network ensembles were trained and validated in 1390 clinical 3D images (DWIs) and tested in 459 images. The lesion core was manually delineated in all the images, serving as "ground truth". Furthermore, we evaluated false positives on extra 499 DWIs of brains with "not visible" lesions. To our best knowledge, this is the largest clinical set ever used. We also tested our trained models on an external population of 140 individuals scanned at the hyperacute stroke phase and 24 h later (STIR dataset[37]). Our results show that our model ensembles are in general comparable to 3D pathwise CNNs (as "DeepMedic") for lesion segmentation, performing superiorly for detection and segmentation of small lesions, with much lower false-positive rate. Our model is generalizable and robust on the external dataset, over the acute phases. Finally, our model is readily and publicly available as a tool with minimal implementation requirements, enabling non-expert users to detect and segment lesions in about one minute in their local computers, with a single command line.

## Methods

**Cohort**. This study included MRIs of patients admitted to the Comprehensive Stroke Center at Johns Hopkins Hospital with the clinical diagnosis of ischemic stroke, between 2009 and 2019 (flowchart for data inclusion in Fig. 1). It utilizes data from an anonymized dataset (IRB00228775), created under the waiver of informed consent because the image is anonymized. We have complied with all relevant ethical regulations and the guidelines of the Johns Hopkins Institutional Review Board, that approved this study (IRB00290649).

From the 2348 DWIs quality-controlled for clinical analysis, 499 images did not show visible lesion. These mostly include transitory ischemic strokes (TIA) or strokes with volume under the image resolution. They are called images with "not visible" lesions and were used to calculate false positives in this study. The other 1849 DWIs showed lesions classified by a neuroradiologist as a result of acute or early subacute ischemic stroke, with no evidence of hemorrhage. The lesion core was defined in DWI, in combination with the apparent diffusion coefficient maps (ADC) by two experienced evaluators and was revised by a neuroradiologist until reaching a final decision by consensus. This manual definition was saved as a binary mask (stroke = 1, brain and background = 0) in the original image space of each subject.

The DWIs with ischemic lesions were randomly split into a training dataset ($n = 1390$), which was used to train and validate all subsequent models, and testing dataset ($n = 459$), which was

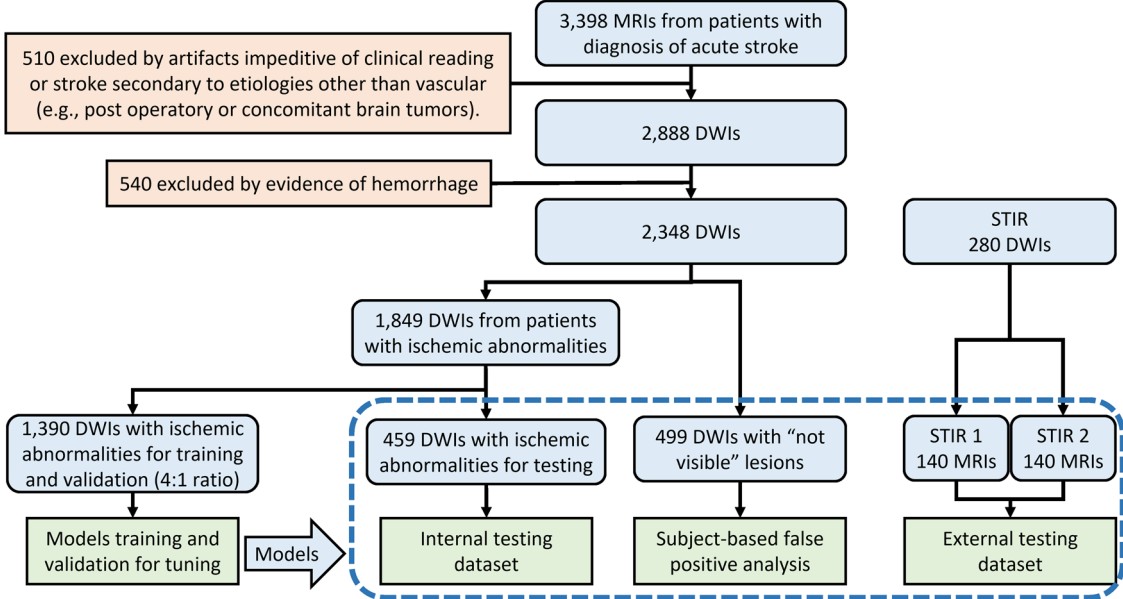

**Fig. 1 Flowchart of data and analysis.** The traced box shows the independent testing samples.

used exclusively for testing models. A second external dataset, STIR, was used to test the generalization of our models in a completely unrelated population (flowchart in Fig. 1). We have complied with all relevant STIR regulations for data usage. From the STIR dataset, we included 140 subjects scanned twice: at the hyperacute stroke event (here called "STIR 1") and up to 24 h later ("STIR 2"). The longitudinal aspect of STIR enables to evaluate the performance of our models according to the stroke's onset.

The demographic, lesion, and scanner profiles for all the datasets are summarized in Table 1. The distribution according to arterial territories (MCA > PCA > VEB > ACA) and the demographic characteristics reflect the general population of stroke patients. MRIs were obtained on seven scanners from four different manufacturers, in different magnetic fields (1.5T (60%) and 3T), with more than a hundred different protocols. The DWIs had high in-plane (axial) resolution ($1 \times 1$ mm, or less), and typical clinical high slice thickness (ranging from 3 to 7 mm). Although a challenge for imaging processing, the technical heterogeneity promotes the potential generalization of the resulting developed tools. The demographics (age, sex, race, time from symptoms onset, and NIHSS) and lesion characteristics (e.g., volume, intensity, location) were balanced between training and testing ischemic sets (Supplementary Table 1).

**Lesion delineation**. The lesion delineations were performed using ROIEditor. The evaluators looked for hyperintensities in DWI and/or hypo intensities (<30% average brain intensity) in ADC. Additional contrasts were used to rule out chronic lesions or microvascular white matter disease. A "seed growing" tool in ROIEditor was often used to achieve a broad segmentation, followed by manual adjustments. The segmentation was performed by two individuals highly experienced in lesion tracing (more than 10 years of experience). In addition, they were trained by detailed instructions and illustrative files, in a subset of 130 cases (randomly selected 10% of the dataset). These cases were then revised by a neuroradiologist, discussed with the evaluators, and (blinded) retraced, revised, and re-analyzed after 2 weeks. The interevaluator index of agreement, Dice score, in this set was $0.76 \pm 0.14$; the intra-evaluator Dice was $0.79 \pm 0.12$. Although this is a satisfactory level of agreement, it demonstrates that human segmentation is not totally reproducible, even when performed by

experienced, trained evaluators. After consistent consensus agreement was achieved in the initial set, the evaluators started working on the whole dataset. The neuroradiologist revised all the segmentation and identified the sub-optimum cases that were subsequently retraced. The segmentations were revised as many times as necessary until reaching a final decision by consensus among all evaluators.

**Image preprocessing**. To convert the images to a common space, where the next preprocessing steps will take place, we: (1) resampled DWI, B0 (the image in the absence of diffusion gradients), and ADC into $1 \times 1 \times 1$ mm$^3$, (2) skull-stripped with an in-house "UNet BrainMask Network", (3) used "in-plane" (IP) linear transformations to map the images to the standard MNI (Montreal Neurological Institute) template, (4) normalized the DWI intensity to reduce the variability and increase the comparability among subjects, (5) "down-sampled" (DS) images to reduce the memory resource requirement in the next steps (e.g., DL networks). Details of these procedures follow.

**Automated skull stripping—BrainMask network**. To decrease computational complexity and time required for skull stripping, we built an "UNet BrainMask Network" using DWI, B0 images, and gold-standard brain masks. The gold standards brain masks were generated as follows: first, all DWI and B0 images were resampled into $1 \times 1 \times 1$ mm$^3$ and skull striped by a level-set algorithm (available under ROIStudio, in MRIStudio[38]), with $W_5 = 1.2$ and 4, respectively (see explanation about the choice of parameters in MRIstudio website[38]). The brain masks were the union of masking on DWI and B0. Then, these brain masks were further manually corrected by our annotators, serving as ground true for the "UNet BrainMask Network".

To train our "UNet BrainMask Network", all images are mapped to MNI and downsampled to $4 \times 4 \times 4$ mm$^3$. The final brain mask inferenced by the network was then post-processed by the closing and the "binary_fill_holes" function from Python scipy module, upsampled to $1 \times 1 \times 1$ mm$^3$, and dilated by one voxel with image smoothing. The Dice agreement between the "gold-standard" brain masks and those obtained with our network was above 99.9%, in an independent test dataset of 713 brains. The average processing time was about 19 s (against 4.3 min taken by the level-set algorithm[38]), making it suitable for large-scale, fast processing.

**Table 1 Population, lesion, and scanner profiles, per dataset.**

| Dataset | Internal dataset | | | | External dataset |
|---|---|---|---|---|---|
| | Training—ischemic | Testing—ischemic | Testing—not visible | STIR 1 | STIR 2 |
| Number of subjects | 1390 | 459 | 499 | 140 | |
| Age in years (median [IQR]) | 62.5 [52, 73] | 62.0 [54, 72] | 61.0 [52, 71] | 73.0 [60.81] | |
| Sex | | | | | |
| Male | 739 (53.2%) | 256 (55.8%) | 251 (50.3%) | 64 (45.7%) | |
| Female | 651 (46.8%) | 203 (44.2%) | 248 (49.7%) | 76 (54.3%) | |
| Race/ethnicity | | | | | |
| African American | 591 (42.5%) | 220 (47.9%) | 236 (47.3%) | 38 (27.1%) | |
| Caucasian | 384 (27.6%) | 138 (30.1%) | 190 (38.1%) | 98 (70.0%) | |
| Asian | 30 (2.2%) | 12 (2.6%) | 9 (1.8%) | 4 (3.9%) | |
| Missing data | 385 (27.7%) | 89 (19.4%) | 64 (12.8%) | 0 (0.0 %) | |
| NIHSS (median [IQR]; missing) | 4.0 [1.0, 8.0]; 681 | 3.0 [1.0, 8.25]; 115 | 1.0 [0.0, 3.0]; 253 | 10.0 [5.5, 16.5]; 1 | |
| Symptoms onset to MRI in hours | | | | | |
| <2 | 65 (4.7%) | 37 (8.1%) | 27 (5.4%) | | |
| 2–6 | 169 (12.2%) | 65 (14.2%) | 63 (12.6%) | | |
| 6–12 | 122 (8.8%) | 106 (23.1%) | 60 (12.0%) | <3 h | ~24 h |
| 12–24 | 349 (25.1%) | 143 (31.2%) | 177 (35.5%) | | |
| >24 | 108 (7.8%) | 12 (2.6%) | 43 (8.6%) | | |
| Missing data | 577 (41.5%) | 96 (20.9%) | 129 (25.9%) | | |
| Lesioned hemisphere | | | | | |
| Left | 627 (45.1%) | 196 (42.7%) | | 80 (57.1%) | |
| Right | 540 (38.9%) | 210 (45.8%) | N.A. | 57 (40.7%) | |
| Bilateral | 223 (16.0%) | 53 (11.5%) | | 3 (2.2%) | |
| Vascular territories | | | | | |
| MCA | 806 (58.0%) | 276 (60.1%) | | 111 (79.3%) | |
| PCA | 258 (18.6%) | 79 (17.2%) | N.A. | 12 (8.6%) | |
| VB | 235 (16.9%) | 67 (14.6%) | | 15 (10.7%) | |
| ACA | 91 (6.5%) | 37 (8.1%) | | 2 (1.4%) | |
| Lesion volume in ml (median [IQR]) | | | | | |
| Any vascular territory | 4.39 [1.05, 21.78] | 4.62 [1.09, 27.84] | | 7.15 [1.44.28.36] | 17.48 [3.21.64.43] |
| MCA | 7.49 [1.78, 28.32] | 6.58 [1.57, 44.23] | | | |
| PCA | 2.53 [0.61, 14.57] | 2.92 [0.64, 15.16] | N.A. | | |
| VB | 1.35 [0.49, 8.94] | 1.74 [0.39, 10.75] | | | |
| ACA | 3.39 [0.78, 8.72] | 2.85 [0.89, 7.39] | | | |
| Lesion contrast in DWI (median [IQR]) | | | | | |
| Any vascular territory | 3.31 [2.28, 4.51] | 3.26 [2.23, 4.41] | | 1.67 [1.24, 2.48] | 3.13 [2.31, 3.99] |
| MCA | 3.28 [2.32, 4.50] | 3.39 [2.47, 4.48] | N.A. | | |
| PCA | 3.20 [2.12, 4.39] | 2.87 [2.13, 4.20] | | | |
| VB | 3.72 [2.49, 5.33] | 3.39 [2.28, 5.16] | | | |
| ACA | 2.76 [2.05, 3.89] | 2.24 [1.55, 3.31] | | | |
| MRI manufacturer* | | | | | |
| Manufacturer 1 (Siemens) | 1207 (86.8%) | 433 (94.3%) | 457 (91.5%) | 0 (0%) | |
| Manufacturer 2 (Phillips) | 13 (0.9%) | 2 (0.4%) | 5 (1%) | 39 (27.9%) | |
| Manufacturer 3 (GE) | 142 (10.2%) | 22 (4.8%) | 29 (5.8%) | 101 (72.1%) | |
| Manufacturer 4 (other) | 28 (2%) | 2 (0.4%) | 8 (1.6%) | 0 (0%) | |
| MRI magnetic field* | | | | | |
| 1.5 T | 930 (66.9%) | 269 (58.6%) | 315 (63.1%) | 104 (74.3%) | |
| 3.0 T | 460 (33.1%) | 190 (41.4%) | 184 (36.9%) | 36 (25.7%) | |
| Voxel size in mm$^3$ (median [IQR]) | | | | | |
| Height/width | 1.20 [0.90, 1.29] | 1.20 [0.60, 1.23] | 1.20 [0.90, 1.30] | 0.88 [0.86, 0.94] | 0.88 [0.86, 0.94] |
| Thickness | 5.00 [4.0, 5.0] | 5.00 [4.0, 5.0] | 5.00 [4.0, 5.0] | 7.00 [4.0, 7.0] | 7.00 [4.0, 7.0] |

ACA, PCA, MCA stand for anterior, posterior, and middle cerebral artery territories, VB stands for vertebro-basilar territory. IQR stands for interquartile range. Statistical significant differences in distributions between testing and training datasets are marked with "*"; P values are in Supplementary Table 3.

**IP-MNI and IP-MNI DS space**. We used sequential linear transformations (3D rigid translation followed by scaling/rotation along $x$ and $y$ axis) to map B0, less affected by the acute stroke, into JHU_MNI_B0[38], a template in MNI space. No rotation was performed along the interslice coordinate ($z$ axis), aiming to preserve the image contrast and the continuity of manual annotations, and to ameliorate issues related to the low resolution and voxel anisotropy. We called this standardized space "in-plane MNI"(IP-MNI). The IP-MNI images were then padded into $192 \times 224 \times 192$ voxels, to facilitates the next step of "down-sampling" (DS) by $(2, 2, 4)$ in $(x, y, z)$ axes and max-pooling operations, which reduces the memory resource requirement in DL networks. The voxel dimension of the downsampled images are $96 \times 112 \times 48$ voxels, which is called IP-MNI DS space.

**DWI intensity normalization**. Intensity normalization increases the comparability between subjects and, as normalizing images to a standardized space, is crucial for diverse image analytical processes. Although the lesion might affect intensity distribution, we assume that the majority of brain voxels are from healthy tissue and can be a good reference for intra- and interindividual comparison. We used bimodal Gaussian function in Eq. (1) to fit the intensity histogram of DWI and cluster two groups of voxels: the "brain tissue" (the highest peak) and "non-brain tissue" (the lowest peak at lowest intensities, composed mostly by cerebrospinal fluid).

$$f(x) = a_1 exp\left(-\left(\frac{x - b_1}{c_1}\right)^2\right) + a_2 exp\left(-\left(\frac{x - b_2}{c_2}\right)^2\right), \quad (1)$$

where $a_i, b_i, c_i$ are the coefficients of the scale, mean, and standard deviation of Gaussian distribution. $a_i, b_i, c_i$ are calculated by least-square fitting the bimodal Gaussian function to the intensity histogram of individual DWI. DWI intensities are normalized to make the "brain tissue" intensity with zero mean and one standard deviation. Supplementary Fig. 1A shows that the DWI intensity distribution of voxels in a brain with ischemic lesions (blue) and in a brain with "not visible" lesion (orange) prior to (left column), and post-to (right column) intensity normalization. We note that the preservation of the minor peak at high intensities in the brain with ischemic lesion indicates the preservation of the lesion contrast after normalization. Supplementary Fig. 1B–D shows the distribution of DWI intensities in groups of images, prior to and post-to intensity normalization. We note that the distributions are much more homogeneous, and the individual variations are smaller after intensity normalization. More importantly, intensity differences between different magnetic fields and scan manufacturers are ameliorated after intensity normalization. Our choice for this intensity normalization approach is, therefore, based on its success to capture the contrast between lesioned voxels and normal tissue voxels, while minimizing variations in intensity range across subjects. The influence of other intensity normalization approaches, from simple $z$ score normalization up to different self-supervised methods[39,40], is subject to further evaluation.

**Lesion segmentation with unsupervised ("classical") methods**. We tried two "classical" methods for lesion segmentation: (1) "t-scores" maps of DWI and ADC for individual brains[14], via comparing them to the set of brains with "not visible" lesions, and (2) ischemic probabilistic maps of abnormal voxels, here called "IS", via the modified c-fuzzy algorithms[15].

*T-score method[14]*. After mapping all the images to a common template (JHU-EVE MNI[38]), we computed the t-score maps of DWI and ADC, $t_{dwi}, t_{adc}$, for each individual s compared to all

controls, as in Nazari-Farsani et al.[14]. The images were smoothed by a Gaussian filter with different full width at half-maximum (FWHM), $W_{fwhm}$. The lesion voxels were detected by thresholding the t-scores maps (i.e., $t_{dwi} > \sigma_{dwi}$, $t_{adc} > \sigma_{adc}$) and thresholding DWI intensities individually (i.e., $t_{id} > \sigma_{id}$). These three thresholds and FWHM were optimized by cross-validation in the training dataset. We searched all parameters' configurations as follows: $W_{fwhm} \in \{2, 3, 4, 5\}$, $\sigma_{dwi} \in \{1, 1.5, 2, 2.5, 3, 3.5\}$, $\sigma_{adc} \in \{1, 1.5, 2, 2.5, 3, 3.5\}$, $\sigma_{id} \in \{1.5, 2, 2.5, 3, 3.5\}$. $\sigma_{dwi}, \sigma_{adc}$, and $\sigma_{id}$ are in $z$ score scale, rather than percentage. $W_{fwhm}$ is in the unit of the pixel. The best model was $W_{fwhm} = 4$, $\sigma_{dwi} = 2$, $\sigma_{adc} = 1$, $\sigma_{id} = 3.5$. The Dice and Net Overlap (defined as[14]) of the top four configurations are summarized in Supplementary Table 2.

*Modified c-fuzzy method[15]*. We first created intensity mean and standard deviation "templates" in IP-MNI space, $I_{s,\mu}(x, y, z)$ and $I_{s,\sigma}(x, y, z)$ for $s = dwi$ and $s = adc$, separately, using the normalized DWI and ADC of cases with "not visible" lesions. We then modified and computed the dissimilarity equation for individual DWI and ADC, $I_s(x, y, z)$ for $s = dwi$ and $s = adc$, compared to $I_{s,\mu}(x, y, z)$ and $I_{s,\sigma}(x, y, z)$ similar as in Guo et al.[15].

$$\triangle I_s(x, y, z) = tanh\left(\frac{I_s(x, y, z) - I_{s,\mu}(x, y, z)}{\alpha_s I_{s,\sigma}(x, y, z)}\right) \quad (2)$$

where $\alpha_s$ is the parameter used for controlling sensitivity. The probability map of abnormal low-intensity (H1) and high-intensity (H2) voxels were computed as:

$$I_{s,H1} = \begin{cases} (-\triangle I_s(x, y, z))^{\lambda_s}, & \triangle I_s(x, y, z) < 0, \\ 0, & x \geq 0 \end{cases} \quad (3)$$

$$I_{s,H2} = \begin{cases} (\triangle I_s(x, y, z))^{\lambda_s}, & \triangle I_s(x, y, z) > 0 \\ 0, & x \leq 0 \end{cases} \quad (4)$$

To generate the probabilistic maps of ischemic voxels, $P_{IS}$, we looked for these voxels to have abnormal high intensity (H2) in DWI and abnormal low intensity (H1) in ADC compared to the templates, and also relatively high intensity in its own individual DWI, is defined as follows:

$$P_{IS}(x, y, z) =$$
$$\begin{cases} I_{dwi,H2}(x, y, z) \times I_{adc,H1}(x, y, z) \times (1 - Q(t_{id}(x, y, z))), & t_{id}(x, y, z) \geq \sigma_{id} \\ 0, & t_{id}(x, y, z) < \sigma_{id} \end{cases},$$
$$(5)$$

where $Q(\cdot)$ is the standard normal distribution Q function, and $t_{id}(x, y, z)$ is the t-score of DWI voxel intensity, compared to the mean and standard deviation of the whole-brain voxels, $\mu_{id}$ and $\sigma_{id}$, in each individual DWI. Like what was described for the t-score method, the DWI and ADC images were smoothed by a Gaussian filter with different FWHM. The six following parameters' combinations were optimized by cross-validation in training dataset: $W_{fwhm} \in \{2, 3, 4, 5\}$, $\alpha_{dwi} \in \{0.5, 1, 1.5, 2, 2.5\}$, $\lambda_{dwi} \in \{2, 3, 4\}$, $\alpha_{adc} \in \{0.5, 1, 1.5, 2, 2.5\}$, $\lambda_{adc} \in \{2, 3, 4\}$, $\sigma_{id} \in \{2, 2.5, 3, 3.5\}$. $\sigma_{id}$ is in $z$ score scale, rather than percentage. $W_{fwhm}$ is in the unit of the pixel. The best parameters' configuration was $W_{fwhm} = 2$, $\alpha_{dwi} = 1.5$, $\lambda_{dwi} = 4$, $\alpha_{adc} = 0.5$, $\lambda_{adc} = 2$, $\sigma_{id} = 2$. The performance in all datasets is summarized in Supplementary Table 3.

As shown in Supplementary Fig. 2, the modified c-fuzzy method performed better than the t-score method for lesion segmentation. In addition, the t-score method performed worse on our testing dataset (Dice about 0.37) than what is described by the developers[14] (Dice about 0.5) Therefore, the classical t-score

method was considered insufficiently efficient to segment lesions in our large and heterogeneous clinical dataset.

## Lesion segmentation with Deep Learning: DAGMNet implementation details

*3D DAGMNet architecture.* The architecture of our proposed 3D DAGMNet, depicted in Fig. 2, is equipped with intraskip connections as UNet3+[24], fused multiscale contextual information block, deep supervision, L1 regularization on final predicts, dual attention gate (DAG)[25–27], self-normalized activation (SeLU)[41], and batch normalization. The details of the important components and training techniques/parameters are outlined in the following subsections.

DAGMNet was intuitively designed to capture semantical features directly from input images at different receptive scale levels, as MNet, and to segment lesions of various volumes with consistent efficiency, via deep supervision at each level. This aims to conquer the drawback that although big lesions can be easily detected at different receptive scale levels in the original generic UNet, small lesions features could be dropped after downsampling pathway/max-pooling layer. Furthermore, The introduction of the interskip connections between layers in UNet structure and intraskip connections as UNet3+ help model share and re-utilize features between different receptive scale levels with lower computational complexity than DenseUNet and UNet++. The final fuse block combines all-scale semantic features (from small lesion volume to large ones) to generate the final predict output. At the end of the fusion block, $L1$-regularization on predicts prevents the networks from false-positive claiming.

To overcome the high variability in lesion volume, shape, and location, DAG was utilized to condition the networks to emphasize the most informative components (both spatial and channel-wise) from encoders' semantic features at each level prior to decoders, increasing the sensitivity to small lesions or lesions with subtle contrast. In DAG, spatial attention gate (sAG) was used to spatially excite the receptive for the most abnormal voxels, like the hyperintensity in DWI or predicts from the third channel ("IS"). On the other hand, the channel attention gate (cAG) was included to excite the most semantical/morphological features associated to ischemic lesions from artifacts.

The inclusion of information from classical methods ("IS") as the third channel aimed to help the networks to focus on abnormal voxels, even if in small clusters (small lesions). In addition, the batch-normalization technique and SeLU activation function aim to self-normalize the networks, avoiding the gradient vanish problem usually faced in complex 3D networks.

*Encoder and decoder.* As shown in Fig. 2, each encoder or decoder convolution block is composed with two consecutive convolution layers with batch normalization and SeLU activation. The number of features at the first level is denoted as "$N_f$". For our proposed model, $N_f$ was 32.

At the encoder part, input images were downsampled by $(2, 2, 2)$ to each receptive scale level and encoded by an encoder convolution block with $N_f = 32$ for each level. At the second, third, and fourth levels, the encoded features from the previous level were concatenated with the features at the present level and furthered encoded by an encoder block with feature number $2 \times N_f$, $4 \times N_f$, and $8 \times N_f$, respectively.

At the decoder part, each level has an decoder convolution block with deep supervision. The feature numbers of decoders at the first, second, third, and fourth levels are $N_f$, $2 \times N_f$, $4 \times N_f$, and $8 \times N_f$. The input for decoders at the first, second, and third-level are features from dual attention gate concatenated with the upsampled decoded features from the second, third, and fourth level, respectively. The decoder at the fourth level only takes the features from dual attention gate as input at the same level.

The final predict is fused by a fuse convolution block at the end. The fuse convolution block takes concatenated features decoded from decoders at each level as inputs. The intraskip connection at each level to fuse convolution block was implemented by transposing up-sampling layers.

*Dual attention gate (DAG).* DAG has two parts: channel attention gate (cAG) and spatial attention gate (sAG). Both have two parts, squeezing and exciting parts as depicted in Fig. 2a.

- cAG block could be considered as a self-attention function intrinsically exciting the most important feature channels. cAG squeeze all spatial features into a channel-wise descriptor by a global average pooling layer and a global max-pooling layer. The most activated local spatial receptive fields would lead to the highest global average/max for its corresponding channel. Then the channel-wise dependencies are calibrated by a fully connected dense layer with sigmoid activation to generate a set of weights of the aggregated spatial information. Each channel will be excited by its corresponding weight post-to the cAG block.

- sAG block could be considered as a self-attention function intrinsically exciting the most important receptive fields. sAG squeezes channel-wise features into spatial statistics with Global Maximum Pooling (GM pooling), Global Average Pooling (GA pooling), and $1 \times 1 \times 1$ convolution layer. Any local spatial semantical features will be excited by a $5 \times 5 \times 5$ convolution layer with SeLU activation to condition network's attention locally. The sum of both excited spatial local features from both inputs is further calibrated by a $1 \times 1 \times 1$ convolution layer with sigmoid activation.

sAG and cAG both take two inputs: "input 1" from low-level encoded features directly from images at the same receptive scale, "input 2" is all accumulated encoded features from all previous levels, including the current scale level.

*SeLU: self-normalization activation function.* Compared to rectified linear unit (ReLU), SeLU activation function was reported to have more power to reduce gradient vanish problem among more complicated network structures. Besides, its self-normalization makes the network learn more from morphological features and be less dependent on image contrast. Hence, we used SeLU activation function in our proposed networks.

*Deep supervision.* Deep supervision is adopted in our model to learn hierarchical representations at each level. Each decoder and the fuse block are fed into a 3-by-3-by-3 convolution layer with sigmoid activation to generate side output, which is supervised by our ground true lesion annotation at its corresponding level.

*Loss function.* To ameliorate the imbalanced voxel classes issue (between the number of lesion and non-lesion voxels) and regularize the false-positive rate predicted by networks, a hybrid loss function described in Eq. (6) was utilized to train our proposed models.

$$L_{final} = L_{fuse} + \sum_{i=1}^{4} L_{i,side}, \qquad (6)$$

where $L_{fuse}$ is the loss function supervised at the final output of the fusion block $X_{fuse}$ and $L_{i,side}$ is the loss function supervised at

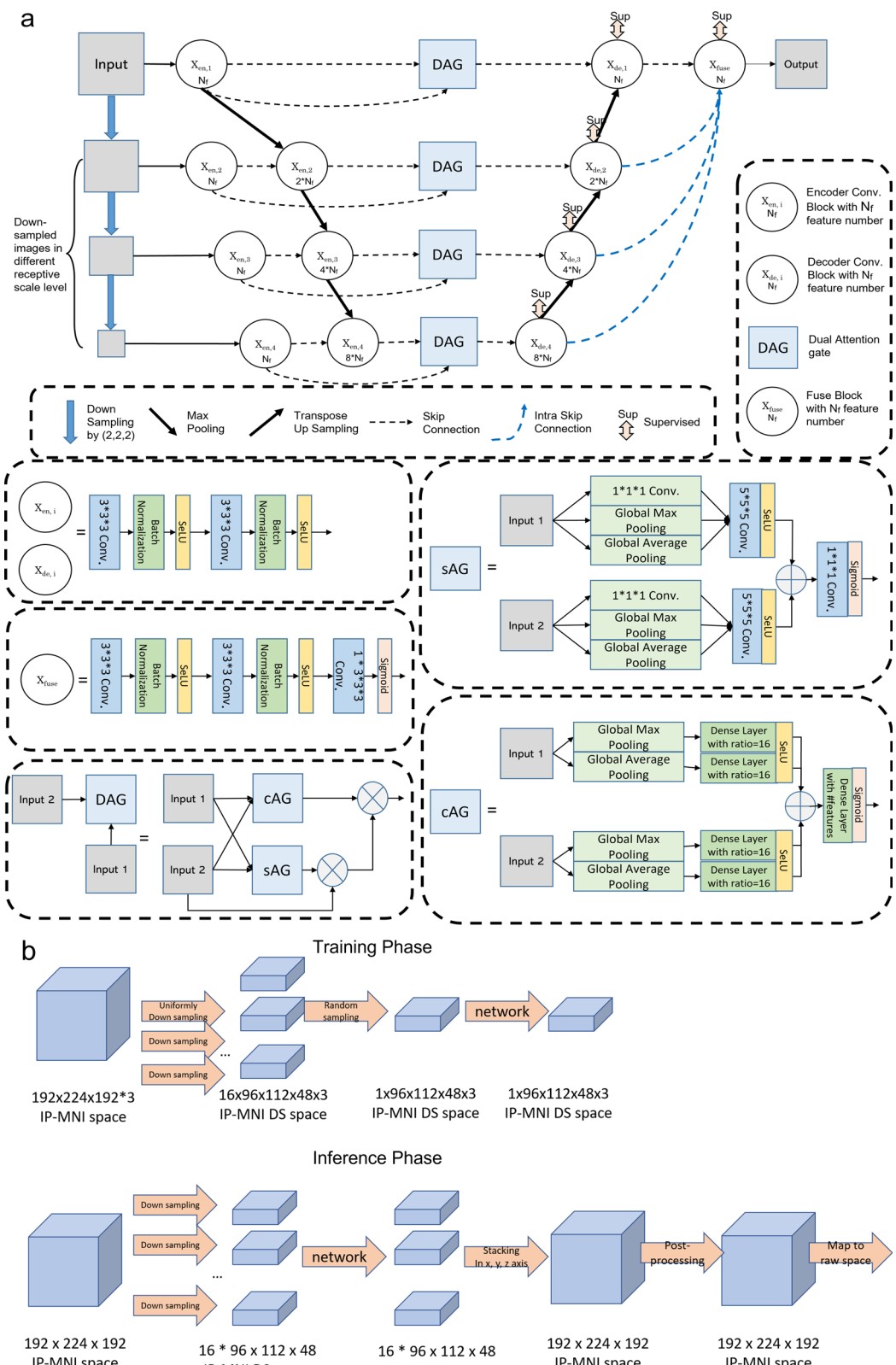

**Fig. 2 Our proposed model for lesion detection and segmentation. a** The architecture of the DAGMNet. In DAG, sAG stands for spatial attention gate and cAG stands for channel attention gate. "$N_f$" denotes the number features ($N_f = 32$ for our final deployed DAGMNet). **b** Flowchart of the input image's dimension for training networks and flowchart of the lesion predict's dimension for inferencing networks in IP-MNI (high-resolution) space.

the side output of the decoders at each level $X_{de,i}$ as follows:

$$L_{fuse} = w_{gds}L_{gds} + w_{bbc}L_{bbc} + w_r L_1 \qquad (7)$$

$$L_{i,side} = w_{gds}L_{gds} + w_{bbc}L_{bbc}, \qquad (8)$$

where $w_{gds} = 1, w_{bbc} = 1, w_r = 1e - 5$ due to hyperparameter tuning, $L_{gds}$ is the generalized dice loss function defined as[42], $L_{bbc}$ is the balanced binary cross-entropy defined as[21], and $L_1(p)$ is the $L_1$ regularization on all predicted voxels defined as

$$L_1 = \sum_{(x,y,z)} |p_{(x,y,z)}|, \qquad (9)$$

where $p_{(x,y,z)}$ is the predicts from networks at $(x, y, z)$ coordinates. The loss function is optimized during training step by ADAM optimizer with learning rate $= 3E - 4$. Learning rate will be factor by 0.5 when loss function is on a plateau over 5 epochs with minimum learning rate $= 1E - 5$.

*The dimension of the inputs and predicts during training and inferencing.* The networks were trained and inferenced in IP-MNI DS space, which is $96 \times 112 \times 48$ voxels. All images (DWI, ADC, IS) in IP-MNI space were downsampled (DS) along $x$, $y$, and $z$ axis with stride of (2, 2, 4) into 16 smaller volumes in $96 \times 112 \times 48 \times 3$ voxels for 3-channel models, as shown in Fig. 2b. The input shape of networks with 3 channels is $96 \times 112 \times 48 \times 3$.

During training step, as shown in Fig. 2b, one of these 16 downsampled volumes are randomly selected to be the inputs of 3-channel networks for each subject in a selected batch (batch size = 4). This re-sampling strategy aims to increase the robustness of our networks to the image's spatial shifting and inhomogeneity. To make efficient backpropagation for training networks, the downsampled volumes were standard normalized to zero mean and unit-variant within the brain mask region for both DWI and ADC channels.

During inferencing step, the 16 lesion predicts from networks in IP-MNI DS space were stacked according to the way their inputs volumes were downsampled in the original coordinates, to construct the final predict in IP-MNI space (Fig. 2c). Then, the lesion mask was "closed" with connectivity 1 voxel. The predicted lesion binary mask (predicted value > 0.5) in IP-MNI space was then mapped back to individual original space. We refined the final prediction by removing clusters with <5 pixels in each slice, which is the smallest size of lesions defined by human evaluators.

*Hyperparameter.* For searching hyperparameters, such as weights for loss functions and networks structures, 20% of subjects from the training dataset were randomly selected as validation dataset, with the same random states for all experiment models. In each experiment, once the loss functions converged in validation dataset along the training epochs (200 epochs at top, early stops at 100 epochs if training and validation loss function converged early), we selected the best model from the snapshot models at every 10 epochs in the validation dataset. We chose maximum training epoch as 200 because most experiments of benchmark models converged after 80 to 120 epochs. For each experiment, we trained the same-type networks independently, with different training set and different resampled validation set, at least twice to check if similar performance would be achieved and avoid overfitting. Once the networks parameters (including weights for loss or regularization, different network layers, depth...etc.) were finalized according to their best dice performance in the validation sets, we used the whole training dataset, including the validation dataset, to train the final deployed models and to make the loss function and dice scores converge to the similar level as the previous experiment. This allowed us to fully use all training dataset and capture the population variation. All Testing

datasets and the STIR datasets were totally unexploited till this step was done.

*System implementation.* All the evaluated methods and models were built with TensorFlow[43] (tensorflow-gpu version is 2.0.0) and Keras[44] (2.3.1) framework on Python 3.6. Imaging processing and analysis were built with Nibabel[45], Scipy[46], Dipy[47], and Scikit-image[48,49]. The experiments run on a machine with an Intel Core (Intel(R) Xeon(R) CPU E5-2620 v4 @ 2.10GHz) with 2 NVIDIA TITAN XP GPUs (with CUDA 10.1). The average inference time over all testing dataset and memory requirements are listed in Table 2.

**Evaluation of efficiency and comparison among methods.** The performances metrics, Dice score, precision, sensitivity, and subject detection rate (SDR), are defined as follows.

$$Dice = \frac{2TP}{2TP + FN + FP} \qquad (10)$$

$$Precision = \frac{TP}{TP + FP} \qquad (11)$$

$$Sensitivity = \frac{TP}{TP + FN} \qquad (12)$$

$$SDR = \frac{\text{the number of the subjects detected with lesions}}{\text{the number of the total subjects in dataset}} \qquad (13)$$

TP, FP, and FN, respectively, denote the number of true positives, false positives, and false negatives in the unit of the voxel. The SDR represents the percentage of the subjects whose lesions are detected. Therefore, high SDR indicates good performance in the datasets of subjects with visible lesions (our testing dataset and STIR) but indicates low performance in the dataset of subjects with "not visible" lesions, because the predicted lesion voxels would be false positives. Hence, in the dataset of subjects with "not visible" lesions, it is false-positive SDR.

All models were compared to the manual delineation of the stroke (used here as "gold-standard") on original raw space. All models were trained, validated, and optimized in the same training dataset, and tested in the independent testing dataset and in the external dataset, STIR. For fair comparison, all models were designed to utilize similar number of trainable parameters, except for DeepMedic, whose default network uses more trainable parameters than all other networks (Table 2).

Besides comparing global performance in the whole brain, we compared models' performance according to lesion volume and location (by vascular territories) and analyzed the Spearman correlation coefficient of performance with the lesion volume and contrast in DWI and ADC. The "scipy.stats" module is used to calculate the Spearman correlation coefficient. The 95%CI for SDR is calculated with $z$ score = 1.96 and the 95% CI for the Spearman correlation coefficient is calculated by Fishers' transformation with $z$ score = 1.96.

The lesion contrast in DWI, $\kappa_{DWI}$, and in ADC, $\kappa_{ADC}$, are defined as the ratio between the average intensity of the lesion voxels and the average intensity of the "normal brain tissue" voxels in DWI and ADC, respectively. High $\kappa_{DWI}$ and low $\kappa_{ADC}$ are expected in acute and subacute lesions which tend to be "bright" in DWI and "dark" in ADC. Furthermore, false-positive analysis was conducted on voxel- and subject-basis among the 499 images with "not visible" lesions.

**Statistical significance testing.** The statistical significance testing was performed by ANOVA test in module "bioinfokit" for continuous data, and by Chi-square test in "scipy.stats.chi2_contingency" module

**Table 2 Performance summary.**

| Metrics | Dataset | DeepMedic | DAGMNet_CH3 | DAGMNet_CH2 | UNet_CH3 | UNet_CH2 | FCN_CH3 | FCN_CH2 |
|---|---|---|---|---|---|---|---|---|
| Dice score | Testing (n = 459) | 0.74 (0.17); 0.79 | 0.76 (0.16); 0.81 | 0.75 (0.17); 0.80 | 0.75 (0.18); 0.81 | 0.74 (0.20); 0.80 | 0.68 (0.20); 0.72 | 0.66 (0.20); 0.71 |
| | STIR 2 (n = 140) | 0.76 (0.18); 0.82 | 0.75 (0.21); 0.82 | 0.75 (0.21); 0.81 | 0.73 (0.24); 0.82 | 0.73 (0.24); 0.82 | 0.70 (0.22); 0.75 | 0.68 (0.24); 0.75 |
| | STIR 1 (n = 140) | 0.55 (0.27); 0.60 | 0.51 (0.30); 0.59 | 0.48 (0.32); 0.58 | 0.49 (0.31); 0.59 | 0.48 (0.32); 0.58 | 0.49 (0.28); 0.55 | 0.44 (0.30); 0.46 |
| | Testing L (n = 163) | 0.85 (0.09); 0.87 | 0.83 (0.10); 0.86 | 0.84 (0.09); 0.86 | 0.85 (0.09); 0.88 | 0.84 (0.10); 0.87 | 0.81 (0.10); 0.84 | 0.80 (0.11); 0.83 |
| | STIR 2 L (n = 76) | 0.84 (0.13); 0.88 | 0.81 (0.18); 0.88 | 0.82 (0.16); 0.89 | 0.81 (0.20); 0.89 | 0.81 (0.21); 0.88 | 0.79 (0.18); 0.84 | 0.77 (0.21); 0.86 |
| | STIR 1 L (n = 50) | 0.67 (0.25); 0.78 | 0.64 (0.28); 0.77 | 0.64 (0.29); 0.79 | 0.59 (0.30); 0.72 | 0.62 (0.30); 0.76 | 0.61 (0.28); 0.74 | 0.59 (0.30); 0.73 |
| | Testing M (n = 144) | 0.74 (0.13); 0.76 | 0.75 (0.14); 0.80 | 0.74 (0.14); 0.77 | 0.76 (0.14); 0.79 | 0.74 (0.16); 0.79 | 0.67 (0.15); 0.71 | 0.66 (0.15); 0.68 |
| | STIR 2 M (n = 43) | 0.73 (0.13); 0.77 | 0.75 (0.13); 0.77 | 0.75 (0.15); 0.78 | 0.72 (0.20); 0.78 | 0.71 (0.22); 0.77 | 0.66 (0.16); 0.70 | 0.66 (0.16); 0.70 |
| | STIR 1 M (n = 51) | 0.53 (0.24); 0.59 | 0.49 (0.28); 0.59 | 0.43 (0.30); 0.50 | 0.45 (0.31); 0.57 | 0.42 (0.30); 0.37 | 0.47 (0.24); 0.53 | 0.39 (0.26); 0.42 |
| | Testing S (n = 152) | 0.63 (0.18); 0.67 | 0.68 (0.19); 0.73* | 0.66 (0.22); 0.72 | 0.65 (0.22); 0.72 | 0.62 (0.25); 0.69 | 0.54 (0.22); 0.58 | 0.51 (0.22); 0.56 |
| | STIR 2 S (n = 21) | 0.52 (0.21); 0.56 | 0.51 (0.25); 0.55 | 0.48 (0.27); 0.51 | 0.49 (0.28); 0.55 | 0.53 (0.26); 0.62 | 0.45 (0.24); 0.48 | 0.40 (0.22); 0.42 |
| | STIR 1 S (n = 39) | 0.43 (0.25); 0.52 | 0.37 (0.29); 0.48 | 0.34 (0.31); 0.29 | 0.41 (0.31); 0.48 | 0.38 (0.32); 0.47 | 0.37 (0.25); 0.42 | 0.32 (0.27); 0.38 |
| Precision | Testing (n = 459) | 0.76 (0.21); 0.82 | 0.83 (0.17); 0.88* | 0.81 (0.18); 0.87 | 0.80 (0.18); 0.86 | 0.81 (0.19); 0.87 | 0.70 (0.22); 0.75 | 0.68 (0.22); 0.73 |
| | STIR 2 (n = 140) | 0.75 (0.19); 0.79 | 0.80 (0.20); 0.87* | 0.78 (0.20); 0.85 | 0.78 (0.21); 0.84 | 0.80 (0.19); 0.85 | 0.72 (0.20); 0.78 | 0.73 (0.20); 0.78 |
| | STIR 1 (n = 140) | 0.62 (0.26); 0.67 | 0.62 (0.31); 0.72 | 0.55 (0.33); 0.64 | 0.65 (0.31); 0.77 | 0.66 (0.32); 0.78 | 0.57 (0.28); 0.65 | 0.57 (0.33); 0.69 |
| Sensitivity | Testing (n = 459) | 0.78* (0.17); 0.82 | 0.73 (0.19); 0.77 | 0.74 (0.21); 0.79 | 0.76 (0.21); 0.83 | 0.71 (0.23); 0.78 | 0.71 (0.21); 0.77 | 0.69 (0.23); 0.76 |
| | STIR 2 (n = 140) | 0.83* (0.21); 0.91* | 0.76 (0.24); 0.85 | 0.78 (0.25); 0.87 | 0.76 (0.28); 0.90 | 0.75 (0.28); 0.88 | 0.74 (0.26); 0.85 | 0.72 (0.27); 0.82 |
| | STIR 1 (n = 140) | 0.59 (0.32); 0.65 | 0.52 (0.33); 0.62 | 0.53 (0.37); 0.65 | 0.48 (0.35); 0.53 | 0.46 (0.35); 0.53 | 0.52 (0.32); 0.61 | 0.43 (0.33); 0.41 |
| Subject detection rate | Testing (n = 459) | 1.00 (0.05); [0.99, 1.00] | 0.99 [0.99, 1.00] | 0.98 [0.97, 1.00] | 0.99 [0.98, 1.00] | 0.98 [0.96, 0.99] | 0.98 [0.97, 0.99] | 0.97 [0.96, 0.99] |
| | STIR 2 (n = 140) | 0.99 (0.08); [0.98, 1.01] | 0.98 (0.14); [0.95, 1.00] | 0.99 (0.12); [0.97, 1.01] | 0.98 (0.14); [0.95, 1.00] | 0.97 (0.17); [0.94, 1.00] | 0.98 (0.14); [0.95, 1.00] | 0.99 (0.12); [0.97, 1.01] |
| | STIR 1 (n = 140) | 0.96 (0.20); [0.92, 0.99] | 0.90 (0.30); [0.85, 0.95] | 0.84 (0.36); [0.78, 0.90] | 0.87 (0.33); [0.82, 0.93] | 0.85 (0.36); [0.79, 0.91] | 0.91 (0.29); [0.86, 0.96] | 0.85 (0.36); [0.79, 0.91] |
| Spearman correlation of dice and lesion volume size | Testing (n = 459) | 0.62 [0.57, 0.68] | 0.44 [0.37, 0.51] | 0.48 [0.41, 0.55] | 0.53 [0.46, 0.59] | 0.53 [0.46, 0.59] | 0.63 [0.57, 0.68] | 0.65 [0.59, 0.70] |
| | STIR 2 (n = 140) | 0.68 [0.58, 0.76] | 0.49 [0.36, 0.61] | 0.54 [0.41, 0.65] | 0.55 [0.42, 0.65] | 0.51 [0.37, 0.62] | 0.60 [0.48, 0.69] | 0.59 [0.48, 0.69] |
| | STIR 1 (n = 140) | 0.42 [0.28, 0.55] | 0.42 [0.27, 0.55] | 0.42 [0.27, 0.55] | 0.30 [0.14, 0.44] | 0.36 [0.21, 0.50] | 0.44 [0.29, 0.56] | 0.42 [0.28, 0.55] |
| Spearman correlation of dice and lesion DWI contrast | Testing (n = 459) | 0.60 [0.54, 0.66] | 0.65 [0.59, 0.70] | 0.61 [0.55, 0.66] | 0.62 [0.56, 0.68] | 0.64 [0.59, 0.69] | 0.64 [0.59, 0.69] | 0.65 [0.59, 0.70] |
| | STIR 2 (n = 140) | 0.45 [0.31, 0.57] | 0.57 [0.44, 0.67] | 0.54 [0.41, 0.65] | 0.54 [0.41, 0.65] | 0.55 [0.42, 0.65] | 0.52 [0.38, 0.63] | 0.56 [0.43, 0.66] |
| | STIR 1 (n = 140) | 0.52 [0.38, 0.63] | 0.56 [0.43, 0.66] | 0.41 [0.26, 0.54] | 0.51 [0.37, 0.62] | 0.40 [0.25, 0.53] | 0.45 [0.30, 0.57] | 0.42 [0.28, 0.55] |

**Table 2 (continued)**

| Metrics | Dataset | DeepMedic | DAGMNet_CH3 | DAGMNet_CH2 | UNet_CH3 | UNet_CH2 | FCN_CH3 | FCN_CH2 |
|---|---|---|---|---|---|---|---|---|
| Spearman correlation of dice and lesion ADC contrast | Testing (n = 459) | −0.33 [−0.41, −0.24] | −0.48 [−0.55, −0.41] | −0.47 [−0.53, −0.39] | −0.44 [−0.51, −0.36] | −0.46 [−0.53, −0.38] | −0.41 [−0.48, −0.33] | −0.40 [−0.48, −0.32] |
|  | STIR 2 (n = 140) | −0.31 [−0.45, −0.15] | −0.37 [−0.51, −0.22] | −0.36 [−0.50, −0.21] | −0.40 [−0.53, −0.25] | −0.42 [−0.55, −0.28] | −0.38 [−0.51, −0.23] | −0.41 [−0.54, −0.26] |
|  | STIR 1 (n = 140) | −0.24 [−0.39, −0.08]+ | −0.30 [−0.44, −0.14] | −0.27 [−0.42, −0.11]+ | −0.29 [−0.44, −0.13] | −0.30 [−0.44, −0.14] | −0.13 [−0.29, 0.03]+ | −0.20 [−0.35, −0.03]+ |
| Spearman correlation of lesion and predict volume size | Testing (n = 459) | 0.97 [0.96, 0.97] | 0.97 [0.97, 0.98] | 0.97 [0.96, 0.97] | 0.97 [0.96, 0.98] | 0.97 [0.96, 0.97] | 0.97 [0.96, 0.97] | 0.97 [0.96, 0.97] |
|  | STIR 2 (n = 140) | 0.97 [0.96, 0.98] | 0.96 [0.94, 0.97] | 0.96 [0.94, 0.97] | 0.93 [0.90, 0.95] | 0.89 [0.86, 0.92] | 0.95 [0.93, 0.96] | 0.94 [0.91, 0.96] |
|  | STIR 1 (n = 140) | 0.87 [0.83, 0.91] | 0.84 [0.79, 0.89] | 0.80 [0.73, 0.85] | 0.81 [0.74, 0.86] | 0.79 [0.72, 0.85] | 0.84 [0.78, 0.88] | 0.79 [0.72, 0.85] |
| Spearman correlation of lesion and predict DWI contrast | Testing (n = 459) | 0.87 [0.85, 0.89] | 0.89 [0.86, 0.90] | 0.88 [0.86, 0.90] | 0.87 [0.84, 0.89] | 0.88 [0.86, 0.90] | 0.86 [0.83, 0.88] | 0.85 [0.82, 0.87] |
|  | STIR 2 (n = 140) | 0.83 [0.77, 0.88] | 0.81 [0.74, 0.86] | 0.85 [0.80, 0.89] | 0.87 [0.82, 0.90] | 0.88 [0.84, 0.91] | 0.83 [0.77, 0.87] | 0.82 [0.76, 0.87] |
|  | STIR 1 (n = 140) | 0.61 [0.50, 0.71] | 0.70 [0.61, 0.78] | 0.59 [0.47, 0.69] | 0.69 [0.58, 0.77] | 0.74 [0.65, 0.81] | 0.59 [0.48, 0.69] | 0.50 [0.36, 0.62] |
| Spearman correlation of lesion and predict ADC contrast | Testing (n = 459) | 0.77 [0.74, 0.81] | 0.84 [0.81, 0.86] | 0.83 [0.80, 0.86] | 0.82 [0.79, 0.85] | 0.83 [0.80, 0.86] | 0.80 [0.77, 0.83] | 0.80 [0.76, 0.83] |
|  | STIR 2 (n = 140) | 0.85 [0.80, 0.89] | 0.86 [0.81, 0.90] | 0.91 [0.87, 0.93] | 0.87 [0.82, 0.90] | 0.93 [0.90, 0.95] | 0.90 [0.86, 0.93] | 0.84 [0.79, 0.88] |
|  | STIR 1 (n = 140) | 0.51 [0.38, 0.63] | 0.58 [0.46, 0.68] | 0.52 [0.38, 0.63] | 0.55 [0.42, 0.66] | 0.57 [0.44, 0.67] | 0.53 [0.39, 0.64] | 0.47 [0.32, 0.59] |
| Median of false positives | Not visible (n = 499) | 14 | 0 | 0 | 0 | 0 | 0 | 0 |
| Number of subjects whose FP > 10 voxels | Not visible (n = 499) | 275 | 132 | 78 | 55 | 36 | 182 | 88 |
| False positive subject detection rate | Not visible (n = 499) | 0.55 (0.50); [0.51, 0.59]* | 0.26 (0.44); [0.23, 0.30] | 0.16 (0.36); [0.12, 0.19] | 0.11 (0.31); [0.08, 0.14] | 0.07 (0.26); [0.05, 0.09] | 0.36 (0.48); [0.32, 0.41] | 0.18 (0.38); [0.14, 0.21] |
| False positive subject detection rate (retrospect evaluation) | Not visible (n = 499) | 0.53 (0.50); [0.48, 0.57]* | 0.24 (0.43); [0.21, 0.28] | 0.14 (0.35); [0.11, 0.17] | 0.10 (0.30); [0.07, 0.12] | 0.06 (0.24); [0.04, 0.08] | 0.34 (0.47); [0.30, 0.38] | 0.15 (0.36); [0.12, 0.18] |
| Number of trainable parameters | All | 24.5 M | 10.7 M | 10.7 M | 10.0 M | 10.0 M | 10.1 M | 10.1M |
| CPU inference time in seconds | Testing (n = 459) | 85.68 | 30.10 (0.52) | 29.09 (0.46) | 19.30 (0.34) | 18.71 (0.37) | 7.15 (0.52) | 6.80 (0.55) |
| GPU inference time in seconds | Testing (n = 459) | 14.97 | 5.91 (0.44) | 4.82 (0.30) | 3.78 (0.18) | 3.59 (0.18) | 2.40 (0.18) | 2.26 (0.18) |

Metrics (dice, precision, sensitivity) are represented as "mean (standard deviation); median"; subject detection rate is represented as "mean (standard deviation). [95% CI]". The correlations are shown as "correlation coefficient; [95% CI]". "+" indicates no significant correlations (P value>1E − 3); all the other correlations were significant with P value≤1E − 3. In dataset column, L = large; M = moderate; S = small lesion groups. The statistical significant difference between DAGMNet_CH3 and DeepMedic is labeled by "*".

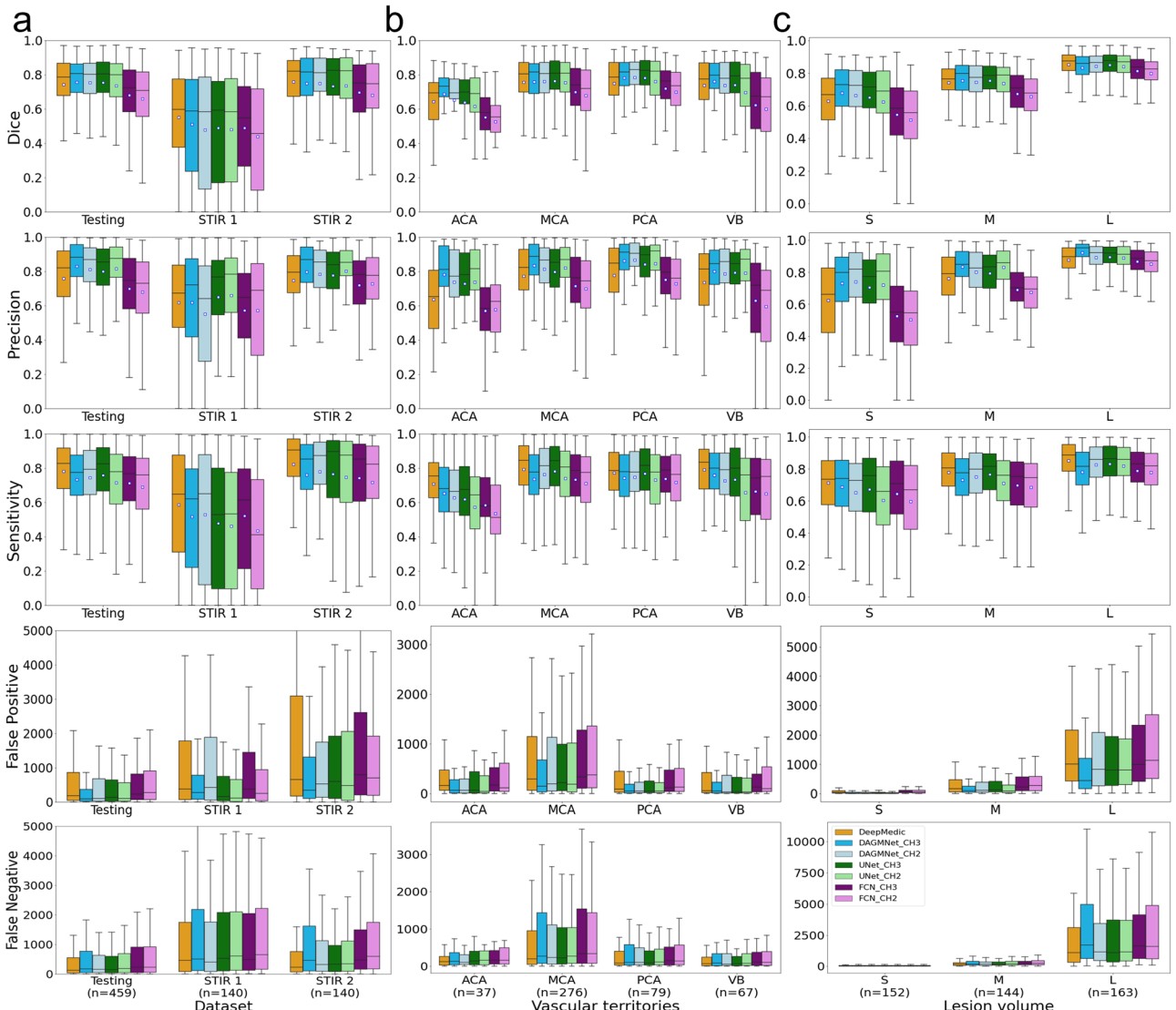

**Fig. 3 Performance of all models (DeeMedic, DAGMNet, UNet, FCN).** The performance is shown according to testing dataset (**a**), vascular territory (**b**), and lesion volume (**c**), S: small, M: medium, L: large. In each Whisker's boxplot, the white square indicates the average; the black trace indicates the median. The whisker is a representation of a multiple (1.5) of the interquartile range (IQR). False-positive and false-negative are in units of voxels, in the original image space. All the sub-figures share the same legend as in sub-figure (**c**).

for categorical data. The statistical significance on the demographic, lesion, and image profiles distribution between training and testing datasets is shown in Supplementary Table 1. For race/ethnicity and MRI manufacturer, we compared the major groups. For symptoms onset to MRI time, we compared groups with onset time < 6 and ≥6 h. The statistical comparison among models' performance in the Testing and STIR dataset is shown in Supplementary Fig. 3. The statistical comparison of models' performance regarding to demographic, lesion, and image profiles is illustrated in Supplementary Fig. 4.

**Reporting summary**. Further information on research design is available in the Nature Research Reporting Summary linked to this article.

## Results

**Lesion segmentation with unsupervised ("classical") methods.** An important question is how the lesion segmentation generated by unsupervised classic methods for abnormal voxel detection compares with DL; and whether the combination of supervised and nonsupervised methods improves the performance. To investigate these questions, we generated ischemic probabilistic maps of abnormal voxels, here called "IS", via the modified c-fuzzy algorithms[15] as detailed in Methods. The best model (Supplementary Table 2) showed modest efficiency on the Testing dataset (Dice: 0.45 ± 0.26) and in STIR 2 (Dice: 0.48 ± 0.28) and was even less efficient in STIR 1 (Dice: 0.23 ± 0.22).

**General comparison of DL models' performance.** Our proposed 3D network, called "DAGMNet" (Fig. 2), was compared with other generic benchmark networks, including FCN, UNet, and DeepMedic. All models were compared to the manual lesion delineation.

In general, "CH3" models that utilize three channel inputs (DWI+ADC+IS, the probabilistic ischemic maps obtained by the modified c-fuzzy method) performed slightly better than "CH2" models (that have only DWI and ADC as inputs). This is particularly noticeable in STIR 1, in a trade-off a slight increase in

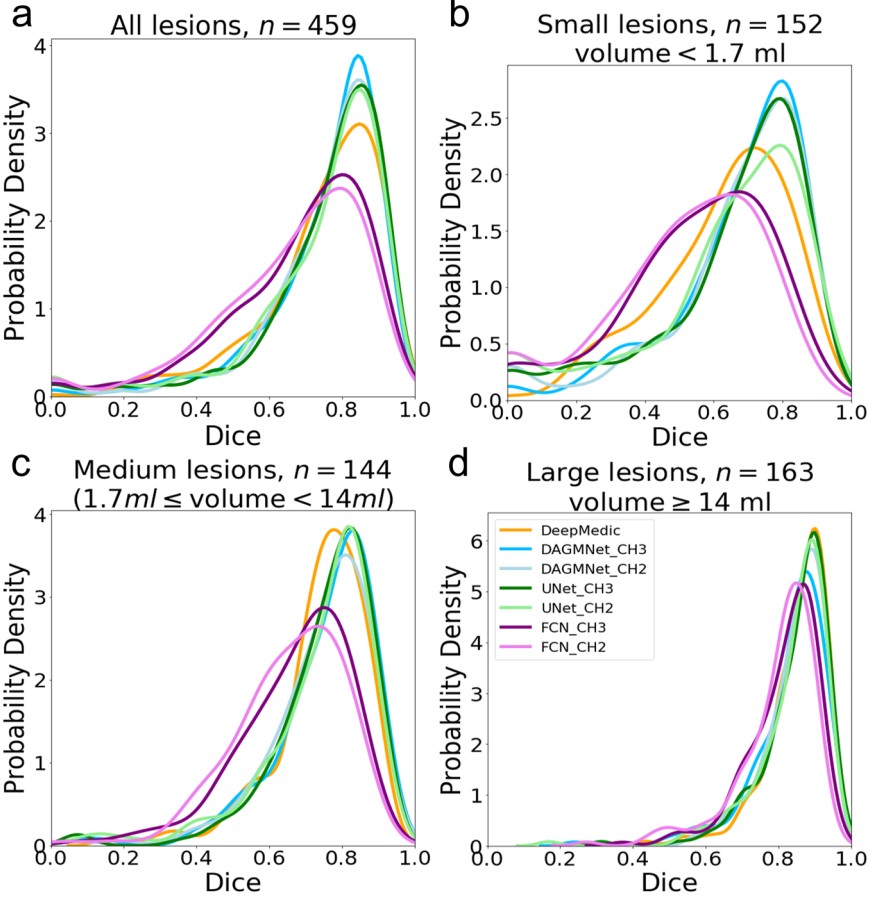

**Fig. 4 The probability density of the Dice score.** The probability density (y axis) of the Dice score of different models in the Testing dataset; (**a**) is for all subjects; (**b**), (**c**), and (**d**) are stratified by the lesion volume. All the sub-figures share the same legends as the sub-figure (**d**).

false positives. It is likely that the extra "IS" channel conditions the network attention to regions of abnormal voxel intensity. This improves the model performance in most of the cases, as the identified abnormal voxels truly correspond to lesions. On the other hand, it might increase the false positives in a minority of cases, in which the abnormal voxels correspond to DWI artifacts.

As depicted in Fig. 3 and Table 2, our proposed model, DAGMNet_CH3, outperforms generic UNet and FCN in all the testing datasets, with higher Dice scores and higher Precision. This is particularly significant by comparing the performances of DAGMNet and FCN in the Testing dataset and STIR 2 (Supplementary Fig. 3). This suggests the robustness and generalization of our proposed model to external data. Compared to DeepMedic, our proposed model shows higher Dice in the testing dataset, comparable Dice in STIR 2, and lower Dice in STIR 1, although all differences are not statistically significant. DAGMNet_CH3 also shows significantly higher precision than DeepMedic in the Testing dataset and STIR 2, but lower sensitivity, with a higher number of false negatives in the Testing set. On the other hand, DeepMedic detected significantly more false positives than all other models, both subject-wise and voxel-wise. For instance, DeepMedic claims false-alarm lesions in more than 55% (twice as our model) of the "not visible" lesion cases.

**Performance according to lesion volume and location.** Taking advantage of our large dataset, we stratified the testing dataset into three comparable sized groups, according to the lesion volumes. These groups contain subjects with lesion volumes here called small, "S" (<1.7 ml, n = 152), medium, "M" (≥1.7 ml

and <14 ml, n = 144), and large, "L" (≥14 ml, n = 163). All the models, except for FCN, perform comparably well in subjects with large lesions, as shown in Figs. 3, 4, and Table 2. Our proposed model and UNet have remarkably higher Dice in subjects with small lesions. Specifically, our proposed model has significantly higher Dice compared to DeepMedic (by average Dice: +0.05, P value: =0.02), UNet_CH2 (by average Dice: +0.06, P value = 0.03), FCN_CH3 (by average Dice: +0.14, P value = 3.9E − 8), and FCN_CH2 (by average Dice: +0.17, P value = 3.0E − 11).

By grouping lesions according to their location in the major vascular territories (anterior, posterior, middle cerebral arteries—ACA, PCA, MCA—and the vertebro-basilar—VB—territory), we observed that all models had their worse performance in ACA and their best performance in MCA (Fig. 3). As shown in Table 1, the MCA lesions are, in average, larger than lesions in other territories, while the ACA territory contains smaller lesions with lower DWI contrast and is more prone to DWI artifacts. Our models outperformed DeepMedic and generic UNet in ACA and VB territories. Last, the trade-off between precision and sensitivity was clear, for all the models, in MCA lesions.

**Performance according to affected hemisphere, population demographics, and scanner profile.** In general, all the models performed consistently, regardless the affected hemisphere, patients' sex and race, symptoms onset to MRI time, scanner manufacturers, and magnetic strengths, as indicated in Fig. 5 and Supplementary Fig. 4. The only exceptions were DeepMedic and

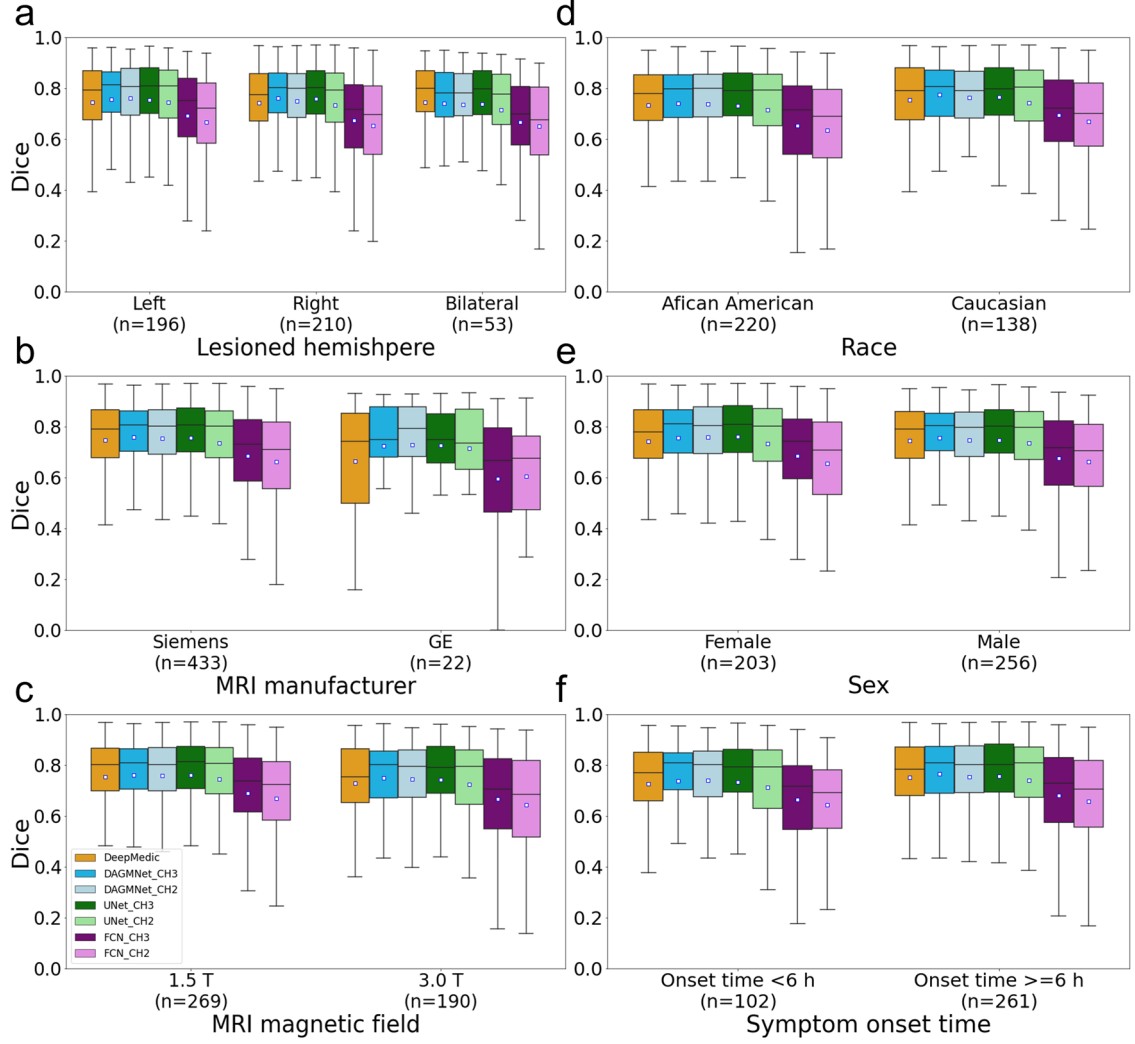

**Fig. 5 Performance of all models (DeeMedic, DAGMNet, UNet, FCN) in the Testing dataset.** The performance is shown according to (**a**) lesion hemisphere, (**b**) scanner manufacturer, (**c**) MRI magnetic field, (**d**) patient's race, (**e**) patient's sex, and (**f**) symptom onset to MRI time. In each Whisker's boxplot, the white square indicates the average; the black trace indicates the median. The whisker is a representation of a multiple (1.5) of the interquartile range (IQR). All the sub-figures share the same legends as sub-figure (**c**).

FCN_CH2, that had significantly lower Dices in Manufacturer 3 (GE), compared to Manufacturer 1 (Siemens) with $P$ values 0.0223 and 0.0413, respectively. However, this is interpreted with caution given the small sample of Manufacturer 3 (GE) in the Testing dataset.

**Relationship of the models' performance with lesion volume and contrast**. The models performance, represented by the Dice agreement between the "human-defined" and the "network-defined" (predicted) lesion positively correlated with the lesion volume and lesion contrast in DWI, and negatively correlated with the lesion contrast in ADC (Fig. 6). This means that, as expected, large and high-contrast lesions are easy to define, both by human and by artificial intelligence. Our proposed model has the lowest correlation between the Dice and lesion volume. Because the Dice score is sensitive to false positives, large lesions tend to have higher Dice than small lesions even for methods with uniform performance across lesion volumes. The decrease of the correlation between Dice and volumes found for our proposed method might result from its superior performance in small lesions. The correlation between the human-defined lesion

volume versus the predicted lesion volume is high for all the models (Table 2 and Fig. 6). Our proposed model showed the highest correlation coefficient between the human-defined lesion contrast versus the predicted lesion contrast, implying the highest agreement with human annotation.

## Discussion

Although several multicenter and clinical trials[2–5] as well as lesion-symptom mapping studies[6] reinforced the need for the quantification of the acute stroke core, there is a gap to be filled by free and accessible technologies that provide fast, accurate, and regional-specific quantification. Commercial platforms[50] are restrictive and do not provide the 3D lesion segmentation required for lesion-based studies or for objective clinical application. Refining methods and disseminating technologies to improve the current lesion estimation will enable better tuning parameters of clinical importance, improving individual classification, and modeling anatomicofunctional relations more reliably and reproducibly. We proposed a DL model for acute ischemic lesion detection and segmentation trained and tested on 2348 clinical DWIs from patients with acute stroke, the largest dataset

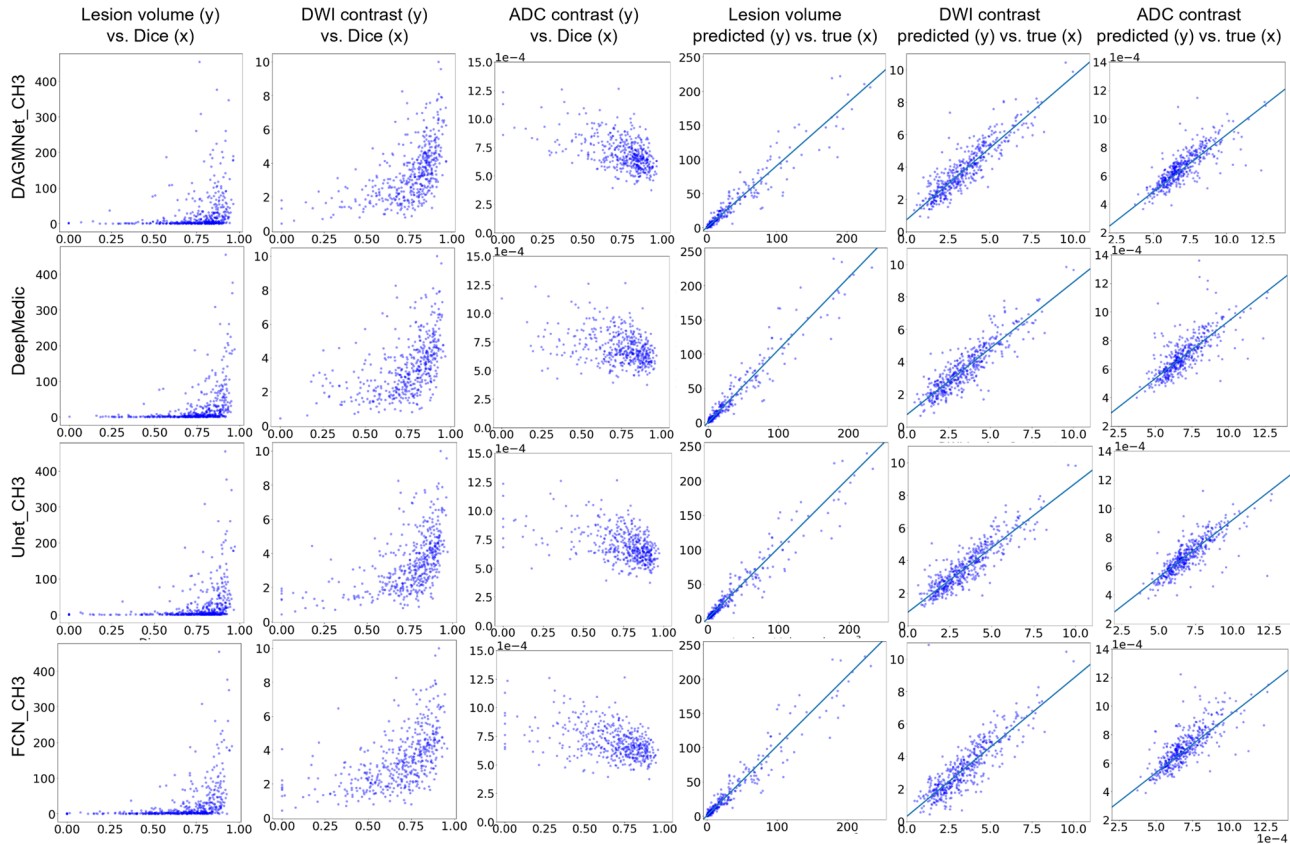

**Fig. 6 Correlation between lesion features and segmentation performance, and correlation between lesion features as segmented manually or automatically.** The first three columns are scatter plots of lesion features (volume, DWI, and ADC contrast) versus Dice score of the different models (rows) in the Testing dataset ($n = 459$). They show how the models perform in lesions of diverse volume and contrast. The last three columns are the metrics of volume, DWI, and ADC of the lesions as traced by human evaluators ("true") vs. as predicted by the different models (rows). Volumes are in ml. ADC values are in $1e-4\ mm^2/s$. Spearman's correlation coefficient and 95% CI are in Table 2. The DWI contrast metrics are defined in "Methods".

employed so far. We further tested this model on an independent, external dataset of 280 images (STIR). The utilization of a large, biologically, and technically heterogeneous dataset, overcomes previous limitations of DL models developed in much smaller datasets, as their questionable generalization potential[36]. The use of real clinical data, which usually imposes extra challenges to image processing, such as low resolution, low signal to noise, and lack information from multiple contrasts, guarantees the robustness of the developed models for large-scale, real scenario applications.

Our proposed model (DAGMNet) showed superior performance than generic models such as UNet and FCN, and DeepMedic, in our independent testing dataset. Our method was significantly superior on segmenting small lesions, a great challenge for previous approaches, while presenting low rate of false-positive detection. This might be a result of the use of large amounts of training data, as well as a contribution of the attention gates and the full utilization of brain morphology by our 3D network, which increases the robustness to technical and biological DWI artifacts. This superiority was evidenced by our method's high performance in vascular territories that contains small lesions and are artifact-prone such as ACA and VB (illustrative examples in Fig. 7). The balanced precision and sensitivity, and the lower dependence on stroke volume and location compared to other methods, make our model well suited not only for lesion segmentation but also for detection, an important component for clinical applications. In addition, our proposed pipeline showed stable performance regardless

the affected hemisphere, population profile and scanner characteristics, other important conditions for large-scale applications.

In the external testing dataset (STIR), our method's performance was again superior or rivaled with that of others. An exception was the lowest Dice achieved in STIR 1, compared to DeepMedic. STIR 1 contains patients with hyperacute strokes (subtle hyperintense or normal intense lesions in DWI). These types of lesions are less frequent in patients with acute and early subacute strokes in our training dataset and particularly challenging for whole-brain-wise networks, compared to local patch-wise networks like DeepMedic. Similarly, our method produced less accurate segmentation in a few cases ($n = 4$) with large, late subacute lesions in STIR 2, that presented subtle intensity contrast in their boundaries (example J in Fig. 7). Note, however, that the advantage of segmenting hyperacute lesions and lesions with subtle contrast was traded-off by the low precision and high false-positive rate of DeepMedic in SITR. The incorporation of the patients with hyperacute strokes in our training dataset, a follow-up local patch-wise segmentation network, and/or using DL techniques like transform learning might ameliorate these issues and are in our future plans. Last, a common source of failure for all methods was errors in skull stripping that increased false positives outside the brain contour. This only occurred in a small number of STIR images ($n = 7$) that presented a high level of background noise due to an outdated acquisition protocol, therefore, is unlikely an issue for future applications.

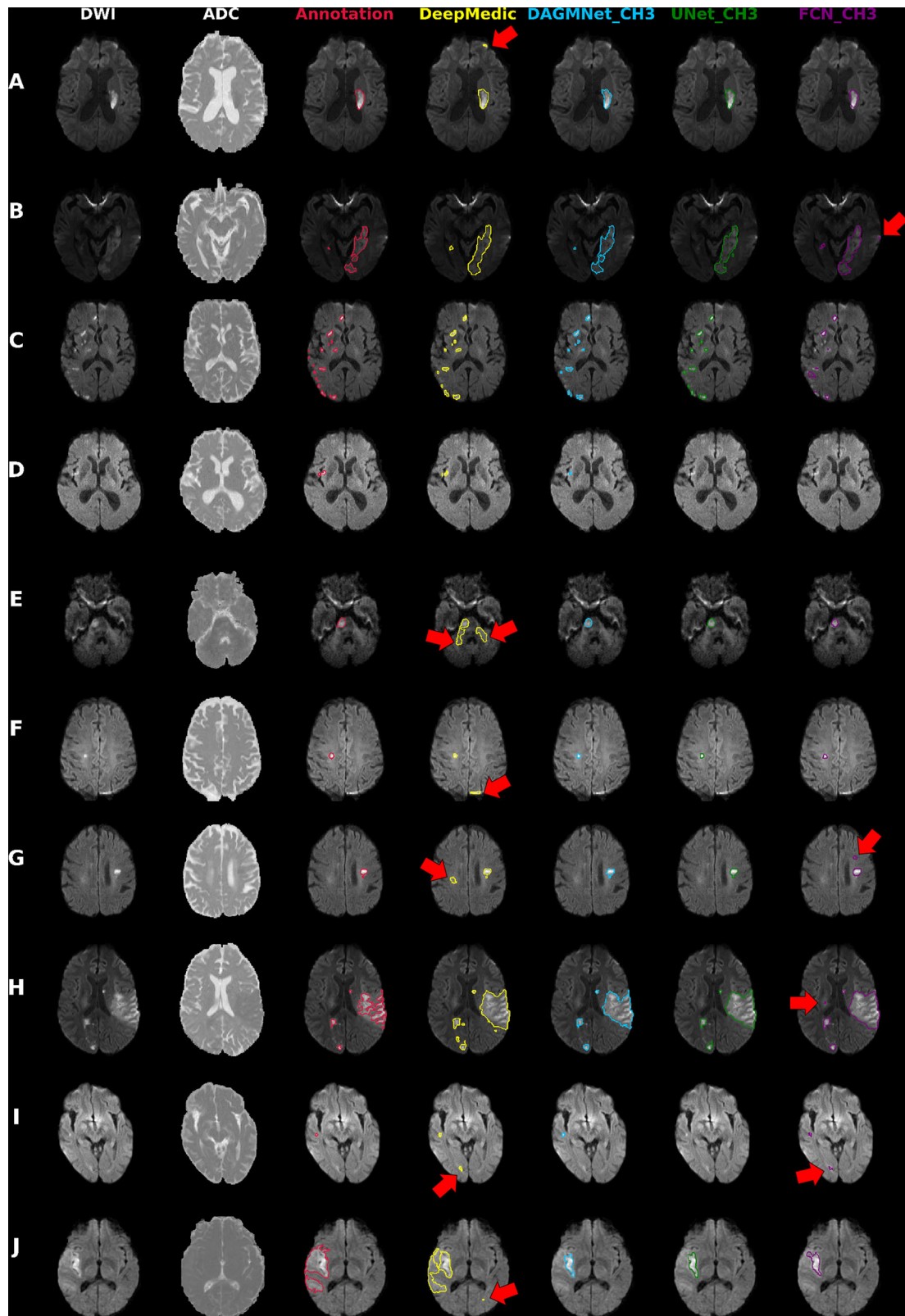

The automated DL models (our proposed DAGMNet, Deep-Medic, and Unet) achieved comparable lesion quantification performance to our experienced evaluators, because their agreement with human delineation was close to the interevaluator agreement (Dice: $0.76 \pm 0.14$, as described in Methods). Among all the tested models, our proposed model showed the highest correlation between the contrast of the predicted lesion and the human-defined lesion, indicating the highest agreement with human evaluation. Still, we noticed circumstantial disagreements of our method with the human evaluators in 24 cases (5% of the testing set), in which the retrospective radiological evaluation favored the results of the automated segmentation, particularly

**Fig. 7 Illustrative cases of models' performances.** Columns from the left to the right: DWI, ADC, and overlays on DWI of the manual delineation (red), DeepMedic Predicts (yellow), DAGMNet_CH3 (our proposed model) predicts (blue), UNet_Ch3 (green), and FCN_CH3 (purple). **A** Typical lesion. **B** Case with inhomogeneity between DWI slices; note the high agreement of our proposed model with the manual annotation. **C** Multifocal lesions. **D** Small cortical lesion, detected exclusively by DeepMedic and our proposed attention model. **E–G** Typical false positives (arrows) of other models (DeepMedic in particular) in areas of: **E** "physiological" high DWI intensities; **F** DWI artifacts in tissue interfaces; in addition to the cortical areas, this was vastly observed in the basal brain, along the sinuses interfaces, and in the plexus choroids; **G** in possible chronic microvascular white matter lesions. **H, I** Cases in which the retrospective analysis favored the automated prediction, rather than the human evaluation for: **H** lesion delineation, and **I** lesion prediction (this case was initially categorized by evaluators as "not visible" lesion, but the small lesion predicted by our model was confirmed by follow-up). **J** Lesion of high-intense core but subtle boundary contrast, which ameliorates the discriminative power of all 3D networks. In this case, the patch-wise DeepMedic had the best agreement with manual annotation in the boundaries at the cost of detecting false positives (arrow).

regarding to the boundaries definition (Fig. 7H). Similarly, by reviewing the "false-positive subjects" (cases in which models' predicted lesions but were initially classified as "not visible" lesions by visual radiological inspection), we identified 29 cases in which the lesion was identified in retrospect, in the light of our method's prediction (example in Fig. 7I). This illustrates the possible advantages of automated methods in reliability and reproducibility.

In addition to be accurate on lesions detection and segmentation, robust to data perturbs (e.g., DWI artifacts, low-resolution sequences, low signal to noise ratio, heterogeneous datasets), and generalizable to external data, our proposed method is fast: it utilizes half of the memory required by its closest competitor, DeepMedic, inferring lesions in half of the time (less than a minute, in CPUs) (see Table 2). All these factors make it potentially suitable for real-time applications. Regardless of all these advantages, a method has no massive utility if only the developers or highly expert analysts are able to use it. Therefore, we made our method publicly accessible on https://www.nitrc.org/projects/ads[51] and ref. [52] and readily useful to run in CPUs with a single command line, and minimal installation requirements. To the best of our knowledge, this study provides the first DL networks for lesion detection and segmentation, trained, and tested over 2000 clinical 3D images, available to users with different levels of access to computational resources and expertise, therefore representing a powerful tool for clinical and translational research.

### Data availability
The source data for the main result figures in the manuscript are available at https://zenodo.org[52]. The original images used for training and testing the models derive from retrospective clinical MRIs and are not publicly available, due to their sensitive information that could compromise research participant privacy. These data can be requested with appropriate ethical approval by contacting Dr. Andreia V. Faria (afaria1@jhmi.edu). The STIR data were used under approval from the STIR steering committee for the current study, and so are not publicly available. These data are however available from the STIR/Vista Investigators upon reasonable request to Dr. Marie Luby (lubym@ninds.nih.gov).

### Code availability
The tool described in this study is publicly available at https://www.nitrc.org/projects/ads[51] and at https://zenodo.org[52].

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

## Acknowledgements

This research was supported in part by the National Institute of Deaf and Communication Disorders, NIDCD, through R01 DC05375, R01 DC015466, P50 DC014664 (A.H., A.V.F.), the National Institute of Biomedical Imaging and Bioengineering, NIBIB, through P41 EB031771 (M.I.M., A.V.F.), and the Department of Neurology, University of Texas at Austin, the National Institute of Neurological Disorders and Stroke, NINDS, National Institutes of Health, NIH (STIR/Vista Imaging Investigators).

## Author contributions

A.V.F. and C.L. conceived and designed the study, analyzed, and interpreted the data, drafted the work. J.H., X.X., S.R., and V.W. analyzed the data. A.E.H. acquired part of the data and substantially revised the draft. M.I.M. revised the draft. The STIR and VISTA investigators provided part of the data.

## Competing interests

M.I.M. owns "AnatomyWorks". This arrangement is managed by Johns Hopkins University in accordance with its conflict-of-interest policies. The remaining authors declare no competing interests.

## Additional information

## The STIR and VISTA Imaging investigators

Max Wintermark[7,8], Steven J. Warach[9], Gregory W. Albers[7], Stephen M. Davis[10], James C. Grotta[11], Werner Hacke[12], Dong-Wha Kang[13], Chelsea Kidwell[14], Walter J. Koroshetz[15], Kennedy R. Lees[16], Michael H. Lev[17], David S. Liebeskind[18], A. Gregory Sorensen[19], Vincent N. Thijs[20], Götz Thomalla[21], Joanna M. Wardlaw[22] & Marie Luby[15]

[7]Radiology, Neuroimaging and Neurointervention, Stanford University, Stanford, CA, USA. [8]Centre Hospitalier Universitaire Vaudois, Lausanne, Switzerland. [9]UT Southwestern Clinical Research Institute of Austin, Department of Neurology and Neurotherapeutics, UT Southwestern Medical Center, Austin, TX, USA. [10]Departments of Medicine and Neurology, Melbourne Brain Centre at the Royal Melbourne Hospital, University of Melbourne, Melbourne, VIC, Australia. [11]Department of Neurology, University of Texas Health Science Center, Houston, TX, USA. [12]Department of Neurology, University of Heidelberg, Heidelberg, Germany. [13]Department of Neurology, Asian Medical Center, University of Ulsan College of Medicine, Ulsan, Korea. [14]Department of Neurology and the Stroke Center, Georgetown University, Washington, USA. [15]National Institute of Neurological Disorders and Stroke (NINDS), National Institutes of Health (NIH), Bethesda, MD, USA. [16]Institute of Cardiovascular and Medical Sciences, University of Glasgow, Western Infirmary, Glasgow, UK. [17]Massachusetts General Hospital and Harvard Medical School, Boston, MA, USA. [18]UCLA Stroke Center, Los Angeles, CA, USA. [19]Siemens Corporate Research, Inc, Princeton, NJ, USA. [20]Laboratory of Neurobiology, Vesalius Research Center, VIB, Experimental Neurology and Leuven Research Institute for Neuroscience and Disease, Department of Neurology, University Hospital Leuven, Leuven, Belgium. [21]University Medical Center Hamburg-Eppendorf, Hamburg, Germany. [22]Brain Research Imaging Centre, Division of Neuroimaging Sciences, Centre for Clinical Brain Sciences, University of Edinburgh, Edinburgh, UK.

