## [Peer Review File · Communications Medicine]

Reviewers' comments:

Reviewer #1 (Remarks to the Author):

This study proposes a fast, public, accessible tool with a new deep learning network (3D DAGMNet) to automatically detect and segment acute ischemic stroke lesion. It combines dual attention gates that fuse channel Attention Gate and spatial Attention Gate, multi-scale learning using down-sampling and up-sampling, self-normalization activation unit and the deep supervision mechanism. It is trained and tested on a large dataset including 2348 DWIs of patients with acute and sub-acute ischemic strokes, and also evaluated on an external public dataset. Experimental results show that the proposed 3D DAGMNet outperforms several state-of-the-arts, particularly in small lesions. The automated lesion quantification of volume and contrast had virtually total agreement with human quantification. It is a very interesting study and provides regular users a public tool. However, this paper is lack of some technique details and explanations, some necessary experiments and discussions, and some statistical analysis. Thus, it cannot be published before revision according to the following concerns.

(1)The authors stated that “The lesion profile (e.g., volume, intensity, location) was balanced between these sets”. However, the authors do not perform statistical test (no P value) to verify this statement.

(2)The non-normal distribution data such as the age, onset to MRI time in Table 1 should be shown in the format: median [IQR] rather than mean and std.

(3)Why not state the exclusion and inclusion criteria of subject selection? Please give a flow chart.

(4)It is better to move the sections: 2.2, 2.4 into the Section: Methods, and the results shown in section 4.5 into the section: Results.

(5)It is not reasonable to stratify all subjects into small and large lesion groups using 1 mL (e.g., 1.5 mL is not considered as large lesion), why not stratify all subjects into three groups: small, moderate and large?

(6)In Table 2, why not give the 95%CI of the correlation coefficient? why no 95%CI for the subject detection rate? is the inference time the average time over all testing subjects? Why no std? and the inference time for DAGMNet_CH3 and DAGMNet_CH2 is the same. Please check. The number of parameters can be shown using $M(1024*1024)$ as the unit.

(7)The reviewer expects more discussions. Why the proposed method performs much worse on >24h subjects than the hyper-acute subjects? Are there any performance differences of the proposed method on left and right lesions? Whether the segmentation performance is associated with the onset-to-MRI time, and sex?

(8)To reduce the memory resource requirement and the computational time, the DWI is down-sampled to $96*112*48$. This step might lose some information, resulting in a poor performance. What's the performance of lesion detection and segmentation when the input size is $192*224*192$? this additional experiment is necessary to verify it is reasonable to down-sample the images.

(9)How about the influence of the proposed Intensity-normalization on the performance? What's

the performance using some other normalization methods such as Z-score and min-max normalization?

(10) In general, in attention gate modules, there is a sigmoid or softmax or normalization operation before element-wise multiplication. Why no such operations in sAG and cAG?

(11) There are no convolution layers in each level of the encoder. I wonder whether adding convolution layers after down-sampling and before inputting into X_{en}, I can improve the performance?

(12) Missing some technical details. What's the number of feature channels (i.e., the number of filters) of each convolution layer? How to decide the weights of each term in the L_{fused} function and why not weights for the L_{fused} and L_{side}? In general, the validation set is used to tune hyper parameters of DL models, however, the authors do not give the details of the validation. Or use 5-fold cross-validation on the whole training set?

(13) Missing legends in several subfigures in Fig. 2. Why no regression line, correlation coefficient and P value in some subfigures in Fig. 3.

(14) Minor concerns: missing the full name of MNI, B0. Two "." at the end of the caption of Table 2. Missing space in "(blue)and". what's the meaning of "fn" in Figure 1? Some Equations are not numbered. Double-check the manuscript.

Reviewer #2 (Remarks to the Author):

This was a clearly written manuscript about automatic acute ischemic stroke (AIS) lesion detection and segmentation on MRIs using a 3D CNN called DAGMNet.

The major finding of this study is developing a 3D CNN model that outperforms generic networks such as FCN, U-Net, and patch-wise approaches like DeepMedic for the AIS lesion segmentation. The authors of this study showed that the application of dual attention gates intra skip connections with the combination of SeLU activation, multi-scale contextual information block, L1-regularization on final predicts, and deep supervision could result in a robust AIS lesion segmentation model. They also claim that their approach surpasses previously developed networks particularly in small lesions owing to several features of their proposed method including 1) lower false-positive rate, 2) balanced precision and sensitivity, and 3) being robustness to data perturbations such as artifacts, low resolution, and technical heterogeneity.

Although the application of ML and DL for lesion and tumor segmentation is not a new topic, as described by the authors, traditional machine learning algorithms that utilize low-level features such as intensity, spatial, and edge information are not optimal to capture the large variation in lesion patterns, especially in real-world low-resolution and noisy clinical data. Convolutional neural networks (CNNs) on the other hand, outperform brain lesion detection and segmentation compared to the traditional ML-based methods. However, medical image segmentation using 2D, 2.5 D, and 3D CNN have their own challenges including lack of generalizability (2D), loss of utilization of full contextual information (2.5 D), and vanishing gradient problem and computational complexity (3D). Accordingly, efficient and accurate segmentation of AIS lesions still remains a challenge and keep

this topic as an interest of clinicians and technicians.

Lack of a large training dataset and accessibility to the real clinical data and inefficiency of synthetic data to represent heterogeneity in real clinical data are obstacles in training generalized models. The utilization of large clinical datasets (more than 2000) including 1390 DWIs for the training and validation and 459 images for the test is one of the strong points of this study. Besides, the authors of this manuscript have given enough details of the work (and publicly available tool), so that the work could be reproducible by other researchers.

Comments:

Introduction:

1.The authors have done a thorough literature review, however, there are two recent studies done at Stanford University that are not opened in the introduction of this manuscript. Due to the relevance of the topic and application of the similar approach (attention gate NN), I strongly recommend the authors discuss these works in the introduction section. Relevant references:

Yu Y, Xie Y, Thamm T, et al. Use of Deep Learning to Predict Final Ischemic Stroke Lesions From Initial Magnetic Resonance Imaging. *JAMA Netw Open*. 2020;3(3):e200772.
doi:10.1001/jamanetworkopen.2020.0772

Y. Yu, Y. Xie, T. Thamm, E. Gong, J. Ouyang, S. Christensen, M.P. Marks, M.G. Lansberg, G.W. Albers, G. Zaharchuk. *American Journal of Neuroradiology* Mar 2021, DOI: 10.3174/ajnr.A7081

2.The authors mentioned a relatively modest training sample size in Zhang et al study. It would give a better intuition to the readers if the authors mention the sample size in the text.

Results:

1.What is the reason for the slightly better performance of the method when it takes 3 channels as input? In other words, which additional information does IS contrast provide to the network that DWI and ADC do not and it leads to the better performance of the method despite the increased false-positive rate?

2.In 2.6, what is the justification of the authors, for setting 1 cm³ as the threshold for dividing the lesions into groups of small and large? How about trying the median size?

3.In 2.7, what is the justification of the authors for the higher dependency of their approach to the lesion contrast compared to the other investigated networks in this study?

Methods:

1.In 4.1 where the authors report the inter-evaluator index of agreement, it would be more

informative to report the intra-evaluator agreement as well.

2. In 4.2, it is not clear which brain masks are served as the gold standard? If automated skull-stripping is one of the sub-goals of this study, it is recommended to add this as a sub-goal in the introduction.

3. In 4.3 please explain the reasons for padding the images.

4. In 4.5.1 “ σ ” is used disconcert. First, σ is referring to the different FWHM sizes. Next, it is referred to configuration parameters (thresholding) ADC and DWI σ_{dwi} , σ_{adc} , and σ_{id} , but it is not clear what are the numbers presenting and what is the unit of the parameters? Are they percentage? If yes, please clarify.

Reviewer #3 (Remarks to the Author):

Strengths are the relatively large sample and the strictly separated training / testing data sets. The results are very good and certainly worth communicating.

To get a better sense of the technical data heterogeneity: What are the specifics of the MRI sequences and scanners (vendors, field strength)?

Lesions can not be classified as “stroke”. Stroke is a clinical event. Please use „diffusion abnormality“ or „ischemic abnormality“ instead of „stroke lesion“.

Discussion (pg. 10/29): “The balanced precision and sensitivity” What does “precision” mean here? Please define.

Discussion (pg. 11/29): “retrospective radiological evaluation favored the results of the automated segmentation” Can the authors give a quantitative summary of those cases?

Reviewer #4 (Remarks to the Author):

In this work, the author proposed a 3D attention-based deep learning model to perform acute ischemic stroke lesion segmentation. The model was trained with a large dataset ($N > 1000$) and the result looks promising. While the proposed idea is interesting and can potentially make a new best benchmark for the acute ischemic stroke lesion segmentation, there are unclear details in the manuscript and there are questions to be addressed. Here are the questions/suggestions:

1. P.1 Abstract: “Our proposed model outperformed generic nets ...”. I understand that authors probably refer “nets” as “neural networks”. For audience coming from non-deep-learning field, they may not be aware of this. Please address this unclarity. Several possible suggestions to replace the word are “neural networks”, “models”, etc.

2. P.3 1. Introduction: “The 3D networks suffer from issues ... no prior information about the

lesion location". Have authors look into the application of a U-net variation (i.e., 3D) in lesion segmentation tasks?

3. P.4 2.1 Cohort: "From the 2,348 DWIs quality-controlled for clinical analysis," It is impressive that the authors have a large and clean ischemic stroke dataset for training deep learning models. In here, it is unclear that whether 2,348 DWIs are obtained from distinct patients (i.e., one patient may have multiple DWIs), please clarify (From Table 1 I found the answer--- 2,348 patients' DWIs; this is unclear in the text).

4. P.5 2.1 Cohort: "... and testing dataset (n=459), ... testing models.". How did the authors come up with this training (N=1,390) and testing (N=459) dataset ratio (~33%), which is uncommon to general machine learning model train-test split (e.g., 80-20, 70-30)?

5. P.6 2.4 Lesion detection and segmentation with deep learning models: "Our proposed 3D network, ... deep supervision." From figure 1, there are many components in the proposed network. (1) How did the authors come up with the network architecture (e.g., use two consecutive conv-batch-*ReLU* layers for X_{en_i} , etc.); (2) this may be out of this paper scope: have authors investigate other possible gains from the attention-based deep learning models? That is, deep learning model is known to be a black-box approach, which is less appealing to real-life medical application (since we cannot explain why the model performance is good). Recently, there are published research work in ischemic stroke applications (e.g., time-since-stroke prediction, etc.) that attempt to interpret deep learning model features. One advantage of the attention-based model is that the model is trained to automatically identify and extract features from the input images that better predict the outcome. Have authors look into model interpretation methods for the proposed model? This may provide additional information to show that the proposed model is superior than other models (e.g., classical models, DeepMedic, etc.)

6. Fig 1A. It is unclear about the input data size. I assume the smaller an input rectangle is, the smaller its input size. Please clarify.

7. P. 7 2.5 General comparison of models' performance: "Our proposed model was compared with ..., and DeepMedic." In the introduction, the authors state that "To our best knowledge, Zhang et al. was the first to reveal the potential of the 3D whole-brain dense networks...". Since Zhang et al. model is the first work and the authors' proposed model is also based on 3D information, why did not the authors include the comparison of Zhang et al. model in this work?

8. P.7 2.5 General comparison of models' performance & table 2: "This is particularly noticeable in STIR1, ... , particularly in STIR." Are the evaluation metrics (e.g., dice coefficient, etc.) statistically significant? (i.e., the performance of DAGMNET_CH3 model is significantly better than other models in term of dice, etc.). There will be quite a lot of p-values for table 2 when authors address this question; one suggestion is to include several p-values in the main context for major model comparisons and put the rest in the supplementary materials.

9. Figure 3. Can the authors increase the font size in the figure? Thank you. Also, please clarify the meaning of the first three columns in Figure 3---Why are there examples that the lesion volume is 0, but the dice coefficient is non-zero?

10. Table 2. It is exciting to see the proposed model perform generally better than other models. Can authors provide an explanation about why the proposed model (DAGMNet_CH3) has lower spearman correlation of dice and lesion volume than the DeepMedic model? (given the fact that the proposed model outperforms the DeepMedic model in all other spearman correlations)

11. 4.6.7 System implementation. "Training our proposed models 200 epochs.... two days." Why 200 epochs were used?

We thank the reviewers for their interest in our manuscript and their thoughtful comments that have allowed us to improve it. Our responses to the reviewers' comments are shown below (the reviewers' original comments are pasted in bold). The changes in the revised manuscript are tracked by the Reviewer comment's number.

Reviewer #1

General comment:

This study proposes a fast, public, accessible tool with a new deep learning network (3D DAGMNet) to automatically detect and segment acute ischemic stroke lesion. It combines dual attention gates that fuse channel Attention Gate and spatial Attention Gate, multi-scale learning using down-sampling and up-sampling, self-normalization activation unit and the deep supervision mechanism. It is trained and tested on a large dataset including 2348 DWIs of patients with acute and sub-acute ischemic strokes, and also evaluated on an external public dataset. Experimental results show that the proposed 3D DAGMNet outperforms several state-of-the-arts, particularly in small lesions. The automated lesion quantification of volume and contrast had virtually total agreement with human quantification. It is a very interesting study and provides regular users a public tool. However, this paper is lack of some technique details and explanations, some necessary experiments and discussions, and some statistical analysis. Thus, it cannot be published before revision according to the following concerns.

R1.1: The authors stated that “The lesion profile (e.g., volume, intensity, location) was balanced between these sets”. However, the authors do not perform statistical test (no P value) to verify this statement.

Thanks for this suggestion. We included the results of statistical tests on the demographic, lesion, and scanner profiles between the training and testing groups (new Table 5 in Methods). We also marked those profiles with statistically significant differences with “*” in Table 1 in the revised manuscript.

As we had stated, there was no significant difference in population or lesion profiles between the training and testing datasets. The only differences that achieved significance level at $p < 0.05$ were the MRI magnetic field (slightly predominance of 1.5T in the training group) and scanner manufacturer (predominance of manufacturer 1 in the training group). Parenthetically, please note that our proposed model, as well as most of the others, did not show difference in performance according to manufacturer or magnetic field. As it would be virtually impossible to balance the groups in each ever aspect, and as the population and lesion characteristics were balanced, we considered the training and testing groups reasonably equitable.

R1.2: The non-normal distribution data such as the age, onset to MRI time in Table 1 should be shown in the format: median [IQR] rather than mean and std.

Thank you for your suggestion. We revised Table 1 accordingly. The median and IQR are now shown in all non-normal distribution data, including age, NIHSS, lesion volume and contrast, and the image voxel size. The symptoms onset to MRI time was acquired in categories and is shown in the original format.

R1.3 Why not state the exclusion and inclusion criteria of subject selection? Please give a flow chart.

A flowchart was added in the revised manuscript (new Figure 1). Briefly, our Stroke Center admits about 500 patients with clinical diagnosis of acute stroke per year (5,000 in the 10 years). About 70% of these patients have MRI with DWI at admission, totalizing the 3,398 MRIs we initially looked at. About 15% of these images were excluded either because they had artifacts impeditive of clinical reading or because the stroke was secondary to etiologies other than vascular (e.g., post operative or concomitant brain tumors). From the cataloged 2,888 cases, 540 (18.7%) had hemorrhagic signals and were not included in this study.

R1.4 It is better to move the sections: 2.2, 2.4 into the Section: Methods, and the results shown in section 4.5 into the section: Results.

We reorganized these sections following your suggestion, thank you. The original section 2.2 (imaging preprocessing) is now moved to the section 4.2 in the Methods. The original section 2.4 (Lesion detection and segmentation with deep learning networks) is combined to subsections 4.7.1, 4.7.3 and 4.7.7 in the Methods.

R1.5: It is not reasonable to stratify all subjects into small and large lesion groups using 1 mL (e.g., 1.5 mL is not considered as large lesion), why not stratify all subjects into three groups: small, moderate and large?

This is a great point, also raised by the Reviewer 2. We now took advantage of our large dataset to capture reasonable thresholds to stratify all the subjects into three comparable sized groups. These groups contain subjects with lesion volumes here called small, “S” (<1.7ml), medium, “M” (≥1.7ml and <14ml), and large, “L” (≥14ml). The numbers of each group are shown in the below table. We reanalyzed the data and revised the manuscript to reflect this new stratification (please see first rows of revised Table 2, Figures 3, 4 and 10, Results section 2.4). Briefly, our proposed model still showed significantly better performance than others in the “small lesion” group.

Groups	Small (S)	Moderate (M)	Large (L)
All data	618	615	616
Training	466	471	453
Testing	152	144	163

R1.6: In Table 2, why not give the 95%CI of the correlation coefficient? why no 95%CI for the subject detection rate? is the inference time the average time over all testing subjects? Why no std? and the inference time for DAGMNet_CH3 and DAGMNet_CH2 is the same. Please check. The number of parameters can be shown using $M(1024*1024)$ as the uint.

We included 95% CI for all the correlation coefficients and SDRs in Table 2. The 95% CI for Spearman’s correlation was calculated via Fisher transformation as in Bishara et al., 2017, Behavior research methods, 49.1: 294-309; the 95% CI for SDR was calculated by z-score=1.96 as in Hazra, 2017, Journal of thoracic disease 9.10: 4125.

The average inference time was originally computed for CH3 in randomly selected 100 testing subjects.

Following the reviewer's suggestion, we re-inferenced all the models over all the testing subjects, under the same computational conditions and updated Table 2. We note that we did not include std for DeepMedic as it runs over a whole batch of subjects by default setting, making hard to track individual inference time.

We re-wrote the number of parameters following the reviewer's suggestions, thank you.

R1.7: The reviewer expects more discussions. Why the proposed method performs much worse on >24h subjects than the hyper-acute subjects? Are there any performance differences of the proposed method on left and right lesions? Whether the segmentation performance is associated with the onset-to-MRI time, and sex?

We thank the reviewer for bringing up these points, which gave us the opportunity to perform further analysis and to expand on these crucial topics.

We now analyze the difference between models' performances according population characteristics (sex, race), and image and lesion profiles (affected hemisphere, MRI magnetic field and scanner manufacturer). As shown in the new Figure 5, our proposed model, and all models in general, perform consistently within all these subgroups. The only significant difference in performance was according to MRI manufacturer: DeepMedic and FCN_CH2 performed worse in Manufacturer 3 (Figure 5). This is interpreted with caution given the small sample of Manufacturer 3 scans in the testing set.

We didn't observe significant difference in models' performance according symptoms onset to MRI time <6h or ≥6h in our testing set. However, in general, all models performed worse (although still rivaling with humans) in hyperacute strokes (<2h, as in STIR1), as compared to acute, late acute, and subacute strokes (STIR2 and internal test set). This might be related to the lowest prevalence of hyperacute cases in the training set (MRIs with strokes younger than 2 hours are relatively infrequent in clinical centers). On the other hand, our method produced relatively less accurate segmentation in four cases of large, late subacute strokes, that presented subtle intensity contrast in their boundaries (one example shown in Figure 4J). This type of lesion is more challenge to whole-brain-wise networks, compared to local patch-wise networks like DeepMedic. Currently, we are trying to ameliorate both issues by adding such cases to our training set and a follow-up patch-wise Siamese segmentation network to "refine" boundaries.

All the new results are included in the Figure 5, Figure 11, Results section 2.5, and discussed in Paragraphs 2 and 3 of the Discussion.

R1.8: To reduce the memory resource requirement and the computational time, the DWI is down-sampled to 96*112*48. This step might lose some information, resulting in a poor performance. What's the performance of lesion detection and segmentation when the input size is 192*224*192? this additional experiment is necessary to verify it is reasonable to down-sample the images.

This is a very good point. We had the same question when started training the networks and did similar experiment as suggested by the reviewer. As the reviewer correctly pointed, using the whole brain high resolution images exhausted the memory resources and greatly increased the training time (as we had to reduce the batch size and feature numbers in convolution layers), which would prevent us to train the network deep enough to capture the lesion features. This raised the concern that we share with the

reviewer: how much, and which kind of information, is lost by downsampling? We note that particularly in this study, the original image resolution is about 1x1 mm in plane, and 4-7 mm inter-slice. When we interpolated to 1x1x1 mm³ in MNI coordinates (to be able to standardize the anatomy for the next steps) we went from low to high resolution slices. However, according to Information Theory principles, the images interpolated to 1x1x1 mm³ do not carry more information than that encoded in their original 4-7 mm slices. Therefore, inter-slice downsampling to 4mm seems a reasonable choice. Regarding the in-plane resolution, it is possible that we lost some information by going from 1mm to 2mm resolution. The question is then how much humans (e.g., trained annotators or radiologists), who made our gold standard, benefited from this information. Based on the illustrative example below, the answer is not much, since images with 1mm or 2mm resolution in plane, produce virtually the same annotation. Furthermore, the way we randomly downsampled to IP-MNI space makes the networks more robust to spatial

inconsistencies, as they are trained with downsampled images with slightly different spatial shifting

Illustrative case resampled to in-plane resolution of 1x1mm² (two panels at left) and 2x2mm², with manual segmentation overlaid in red. In both cases, the measured lesion volume (4.3ml), mean intensity (DWI normalized) and standard deviation (4.9 ± 2.3) were the same.

We note that the above arguments about the cost/benefit of downsampling are very particular to this study, which uses low resolution clinical images as inputs and aims to detect a “high effect size” abnormality (i.e., the lesion contrast boundary). We share the reviewer’s opinion that this is an important issue and must be considered in each specific case.

R1.9: How about the influence of the proposed Intensity-normalization on the performance? What’s the performance using some other normalization methods such as Z-score and min-max normalization?

This is again a great point. Intensity-normalization is indeed a crucial point for DL models and imaging analysis in general. Our choice for intensity normalization was based on its success to capture the contrast between lesioned voxels and normal tissue voxels, while minimizing variations in DWI intensity range across subjects (as indicated by the consistent post-normalization intensity range over manufacturer, magnetic field etc, shown in the Figure 8 in the Methods). Nevertheless, we do believe that other normalization approaches, such as the z-score mentioned by the reviewer, can result in similar performance, as long as the networks are sufficiently improved and trained (e.g., using extra convolution layers and more parameter tuning/searching steps during training phase).

To investigate the influence of the intensity normalization, we used images normalized by three different

methods: 1) standard z-score normalization on whole images ('StandardNorm'), 2) standard z-score normalization on brain-masked region only ('BrainMaskStandardNorm'), 3) Max-Min normalization ('MaxMinNorm'), compared to our proposed normalization method ('ProposedNorm'), keeping all other procedures for training the network, inferencing predicts, and the post-processing, unaltered. We trained and 5-fold cross-validated all proposed methods with UNet_Ch2 (for the easiness of training). All methods converged to the same level in validation Dice and loss function within 100 epochs as in the figures below.

As shown by the Dice scores below (mean(std); median) on the Testing dataset, all the three intensity normalization methods tested resulted in inferior performance compared to our proposed normalization.

Model	ProposedNorm	StandardNorm	BrainMaskStandardNorm	MaxMinNorm
Dataset				
Testing	0.74(0.19);0.80	0.57(0.24);0.64	0.59(0.30);0.70	0.53(0.25);0.60
STIR 2	0.75(0.22);0.82	0.59(0.25);0.65	0.61(0.31);0.74	0.55(0.28);0.62
STIR 1	0.50(0.32);0.59	0.30(0.29);0.24	0.17(0.27);0.00	0.26(0.28);0.13

Dice on Testing dataset

While we believe that the large variability in the intensity scale across DWIs and populations is a great challenge for simple normalization methods, we also believe that more complex training (better loss function or parameter searching/more convolution layers) might alleviate this issue. In addition, other studies have been using different self-supervised networks to reduce the bias and noise at the preprocessing steps [i.e. Alejandro et al. 2020 <https://arxiv.org/abs/2006.13944> and Fadnavis et al. 2020 <https://arxiv.org/abs/2011.01355>], which can be a feasible future approach. Although testing and optimizing pre-processing steps, as intensity normalization, is not in the scope of this study, we recognize the importance of the point raised by the reviewer and included the following sentence in section 4.5:

“Our choice for this intensity normalization approach is, therefore, based on its success to capture the contrast between lesioned voxels and normal tissue voxels, while minimizing variations in intensity range across subjects. The influence of other intensity normalization approaches, from simple z-score normalization up to different networks [i.e. Alejandro et al. 2020 <https://arxiv.org/abs/2006.13944>] is subject to further evaluation”.

R1.10: In general, in attention gate modules, there is a sigmoid or softmax or normalization operation

before element-wise multiplication. Why no such operations in sAG and cAG?

We apologize for the missing information: we do have the sigmoid operation at sAG and cAG, but they were missed in the Figure 2, which is now updated accordingly. Furthermore, we rephrased the dual attention gate subsection (section 4.7.3 in the revised manuscript) including details missed in the original draft.

R1.11: There are no convolution layers in each level of the encoder. I wonder whether adding convolution layers after down-sampling and before inputting into $X_{en,i}$ can improve the performance?

There are some convolution layers at each level after down-sampling, prior to all DAG's (red boxes in the figure below). They are used to capture lower-level features of the inputs at each down-sample domain. Please let us know if we misunderstood your question and you are suggesting something else.

R1.12: Missing some technical details. What's the number of feature channels (i.e., the number of filters) of each convolution layer? How to decide the weights of each term in the Lfused function and why not weights for the Lfused and Lside? In general, the validation set is used to tune hyper parameters of DL models, however, the authors do not give the details of the validation. Or use 5-fold cross-validation on the whole training set?

We again apologize for the missing information. We include these details and further clarifications in sections 4.7.6 and 4.7.8 of the revised manuscript. The edits are as below:

The number of features ("fn", as defined in Figure 2 of the revised manuscript), was chosen to make trainable parameters comparable among models. For DeepMedic, we used their default model without any change. For UNet and FCN architectures, we used the original structures proposed in their papers, and adapted the input size as the table below, so they used similar number of trainable parameters as ours.

Model	DAGMNetCH3	DAGMNetCH2	UNetCH3	UNetCH2	FCNCH3	FCNCH2
Fn	32	32	40	40	14	14

The weight of each term in L_{fuse} was chosen to make each loss function term converge to similar scale during training steps. We did try different weight for L_1 regularization term ($L_1 = 1e-4, 1e-5, \text{ and } 1e-6$). $L_1=1e-4$ suppressed too many predicts, generating a lot of zeros and increasing false negatives. $L_1=1e-6$ did not have enough regularization after training certain epochs, being too small compared to other loss function terms (less $1e-2$ scale). L_{gds} and L_{bbc} were initially weighted differently, which did not result in noticeable difference. When models' Dice approached inter-evaluator's Dice, further tune seemed unnecessary. The same reasoning was true for L_{fused} and L_{side} : other trials (like $L_{fused}=10, 5 \text{ or } 2$; and $L_{side}=1$) did not result in noticeable differences, and we eventually opted by the simplest option, keeping them the same.

Validation: As training networks took over 1-2 days, we did not use cross-validation in each experiment for the sake of time, which is a common practice in this scenario. Instead, in each experiment, 20% of subjects from the training dataset were randomly selected as validation dataset, with the same random states for all experiment models, for searching parameters such as weights, networks structures etc in each experiment. Once the loss functions converged in validation dataset along the training epochs (200 epochs at top, early stops at 100 epochs if training and validation loss function converged early), we selected the best model (snapshot models every 10 epochs) in the validation dataset. For each experiment, we trained the same-type networks independently, with different training set and different resampled validation sets, at least twice to check if similar performance would be achieved and to avoid overfitting. Once the networks parameters (including weights for loss or regularization, different network layers, depth...etc) were finalized according to their best performance in the validation sets, we used the whole training dataset, including validation, to train the final deployed model and to make the loss function and dice scores converge as in the previous training dataset (with a few validation samples aside for early stop as suggested by Amari, Shunichi, et al. "Asymptotic statistical theory of overtraining and cross-validation." IEEE Transactions on Neural Networks 8.5 (1997): 985-996). This allowed us to fully use the training dataset and better capture the population variation. The Testing dataset and STIR were totally unexploited till this step.

As a matter of fact, we did a 5-fold cross validation experiment on UNet_Ch2 (for the easiness of training. Each fold took about 1.5 weeks to train; the whole cross-validation experiment took about 3 weeks on 4 GPU's.) As shown by the Dice scores below (average(std); median), there was no significant differences between the Testing Dice results of 5-fold cross-validated models (left) or our current validation approach deployed model (right). Actually, the Dice scores were virtually the same.

Dataset		UNet_CH2
Testing	0.74(0.19);0.80	0.74(0.20);0.80
STIR 2	0.75(0.22);0.82	0.73(0.24);0.82
STIR 1	0.50(0.32);0.59	0.48(0.32);0.58

Dice on Testing dataset

Therefore, although a fully 5-fold or 10-fold cross validation could result in a network of a slightly different performance, to fully brutal search large parameters configurations would be time-prohibitive with the available computational resources. Ultimately, we reinforce that rather than arguing that we created the

best possible model, we keep our contribution restricted on offering a reasonably efficient pre-trained model, accessible to non-expert researchers, able to support large scale reproducible data analysis.

R1.13: missing legends in several subfigures in Fig. 2. Why no regression line, correlation coefficient and P value in some subfigures in Fig. 3.

All the panels in Figure 2 (of the original draft, now Figure 3) share the same legends, at the bottom right. This is now explained in the figure caption, thank you for noticing this missing information.

For Figure 3 (of the original draft, now Figure 6), the correlation between lesion size, DWI and ADC contrast with Dice is not expected to be linear. Hence, drawing straight fitting line would not provide the same useful information as the fitting line on the correlation between ground trues and predicts of volume, DWI and ADC contrast.

All the correlation coefficients and 95% CI (recommended in your comment #6) are in Table 2, as it would be hard to include them in the figure with a readable font size.

R1.14: Minor concerns: missing the full name of MNI, B0. Two “.” at the end of the caption of Table 2. Missing space in “(blue)and”. what’s the meaning of “fn” in Figure 1? Some Equations are not numbered. Double-check the manuscript.

We appreciate your careful revision. The revised draft was double-checked and proof-read at our best. MNI is defined as “Montreal Neurological Institute” at the section 4.2 in the revised draft and B0 is now defined as “the image in the absence of diffusion gradients” at the section 4.2 of the revised draft. Fn means the number of features; it’s added at the Methods and specified in Figure 2 of the revised manuscript.

All notation definitions were checked and added, if missing, in the revised version.

All equations were numbered properly as well.

Reviewer #2

General comments:

This was a clearly written manuscript about automatic acute ischemic stroke (AIS) lesion detection and segmentation on MRIs using a 3D CNN called DAGMNet.

The major finding of this study is developing a 3D CNN model that outperforms generic networks such as FCN, U-Net, and patch-wise approaches like DeepMedic for the AIS lesion segmentation. The authors of this study showed that the application of dual attention gates intra skip connections with the combination of SeLU activation, multi-scale contextual information block, L1-regularization on final predicts, and deep supervision could result in a robust AIS lesion segmentation model. They also claim that their approach surpasses previously developed networks particularly in small lesions owing to several features of their proposed method including 1) lower false-positive rate, 2) balanced precision and sensitivity, and 3) being robustness to data perturbs such as artifacts, low resolution, and technical heterogeneity.

Although the application of ML and DL for lesion and tumor segmentation is not a new topic, as described by the authors, traditional machine learning algorithms that utilize low-level features such as intensity, spatial, and edge information are not optimal to capture the large variation in lesion patterns, especially in real-world low-resolution and noisy clinical data. Convolutional neural networks (CNNs) on the other hand, outperform brain lesion detection and segmentation compared to the traditional ML-based methods. However, medical image segmentation using 2D, 2.5 D, and 3D CNN have their own challenges including lack of generalizability (2D), loss of utilization of full contextual information (2.5 D), and vanishing gradient problem and computational complexity (3D). Accordingly, efficient and accurate segmentation of AIS lesions still remains a challenge and keep this topic as an interest of clinicians and technicians.

Lack of a large training dataset and accessibility to the real clinical data and inefficiency of synthetic data to represent heterogeneity in real clinical data are obstacles in training generalized models. The utilization of large clinical datasets (more than 2000) including 1390 DWIs for the training and validation and 459 images for the test is one of the strong points of this study. Besides, the authors of this manuscript have given enough details of the work (and publicly available tool), so that the work could be reproducible by other researchers.

R2.1 The authors have done a thorough literature review, however, there are two recent studies done at Stanford University that are not opened in the introduction of this manuscript. Due to the relevance of the topic and application of the similar approach (attention gate NN), I strongly recommend the authors discuss these works in the introduction section. Relevant references:

Yu Y, Xie Y, Thamm T, et al. Use of Deep Learning to Predict Final Ischemic Stroke Lesions From Initial Magnetic Resonance Imaging. *JAMA Netw Open*. 2020;3(3):e200772.

doi:10.1001/jamanetworkopen.2020.0772

Y. Yu, Y. Xie, T. Thamm, E. Gong, J. Ouyang, S. Christensen, M.P. Marks, M.G. Lansberg, G.W. Albers, G. Zaharchuk. *American Journal of Neuroradiology* Mar 2021, DOI: 10.3174/ajnr.A7081

Thank you for this suggestion. We agree these studies are relevant to our work and now briefly discuss them in the Introduction (3rd paragraph).

R2.2 The authors mentioned a relatively modest training sample size in Zhang et al study. It would give a better intuition to the readers if the authors mention the sample size in the text.

Thank you for identifying this missing information. In the referred study, a total of 242 low-resolution images were used for training, validating, and testing the models; plus 36 high-resolution images, downsampled to low-resolution, to test for generalization. This is now included in the fourth paragraph in the introduction.

R2.3 What is the reason for the slightly better performance of the method when it takes 3 channels as input? In other words, which additional information does IS contrast provide to the network that DWI and ADC do not and it leads to the better performance of the method despite the increased false-positive rate?

This is a very interesting question, that we asked ourselves when interpreting the results. The third channel, IS, gives the networks extra information on the spatial location of the voxels with “abnormal intensities” (lesioned or artifact voxels). This could condition the networks to focus on these suspicious regions, rather than capturing features from scratching the whole brain. Take the spatial attention gates, for example. The third channel, IS, would have higher values after spatial global pooling and the networks would pay more attention on these corresponding spatial channels. On the other hand, the IS predicts from classical methods also contain false positives, particularly in artefact-prone regions. In these cases, the networks are expected to identify the morphological feature differences between lesions and artefact. Inevitably, this might introduce more false positives in the 3-channel models. As this is a very interesting point, we added the following sentence the subsection 2.3, “General comparison of DL models’ performance”.

“It is likely that the extra “IS” channel conditions the network attention to regions of abnormal voxel intensity. This improves the model performance in most of the cases, as the identified abnormal voxels truly correspond to lesions. On the other hand, it might increase the false positives in a minority of cases, in which the abnormal voxels correspond to DWI artefacts”.

R2.4 In 2.6, what is the justification of the authors, for setting 1 cm³ as the threshold for dividing the lesions into groups of small and large? How about trying the median size?

This is a great point, also raised by Reviewer 1 Comment 5. We now take advantage of our large dataset to capture reasonable thresholds to stratify the subjects into three comparable sized groups. These groups contain subjects with lesion volumes here called small, “S” (<1.7ml), medium, “M” (≥1.7ml and <14ml), and large, “L” (≥14ml). The numbers of each group are shown below. We reanalyzed the data and revised the manuscript to reflect this new stratification (please see first rows of revised Table 2, Figures 3, 4 and 10, and Results section 2.4). Briefly, our proposed model still showed significantly better performance than others in the “small lesion” group.

	Small (S)	Moderate (M)	Large (L)
All data	618	615	616
Training	466	471	453
Testing	152	144	163

R2.5 In 2.7, what is the justification of the authors for the higher dependency of their approach to the lesion contrast compared to the other investigated networks in this study?

The sentence “As shown by the correlation coefficients in Table 2, the performance of our model (DAGMNet_CH3) is more dependent on the lesion contrast, and less dependent on the lesion volume than all the other model’s performance” is inaccurate and was removed. We apologize for the misconstruction. This section meant to show that human-defined lesion features strongly correlated with features of the lesion as predicted by any of the models. We also want to mention that our proposed model showed the strongest correlation, indicating the highest agreement with human delineation.

R2.6 In 4.1 where the authors report the inter-evaluator index of agreement, it would be more informative to report the intra-evaluator agreement as well.

We included the intra-evaluator agreement as well (0.79 ± 0.12) at section 4.1 in the revised draft, thank you for your suggestion.

R2.7 In 4.2, it is not clear which brain masks are served as the gold standard? If automated skull-stripping is one of the sub-goals of this study, it is recommended to add this as a sub-goal in the introduction.

The gold standard brain masks for training and testing our automated skull-stripping algorithm were generated by the method in our reference [40], followed by manual correction. We re-wrote section 4.2 (now it is the section 4.3 in the revised draft) to make it clearer, as well as to clarify other steps of the skull stripping process. Although our automated skull-stripping algorithm was highly efficient, it is not a goal of this study, and we prefer to keep the details in the Methods.

R2.8 In 4.3 please explain the reasons for padding the images.

An even image size facilitates the “down-sample” required in further steps (otherwise, extra padding or cropping would be needed during “up-sampling/transposed convolution” step). We padded images to the closest even number with factors of power of 2, such as $192 = 3 \times 2^6$.

R2.9 In 4.5.1 “ σ ” is used disconcert. First, σ is referring to the different FWHM sizes. Next, it is referred to configuration parameters (thresholding) ADC and DWI σ_{dwi} , σ_{adc} , and σ_{id} , but it is not clear what are the numbers presenting and what is the unit of the parameters? Are they percentage? If yes, please clarify.

Thank you for catching up this inconsistency and the missing information. We assume that the reviewer is referring to “sigma” (“ σ ” is probably a format issue – please let us know if we misunderstood it). For consistency, we changed all the notation for FWHM size to “*fwhm*”. The “sigma” is threshold in the z-score scale, not percentage. This information is now added at the section 4.6.1 and 4.6.2 of the revised manuscript, as below.

σ_{dwi} , σ_{adc} , and σ_{id} are in z-score scale, rather than percentage.

σ_{id} is in z-score scale, :

Reviewer #3

General comment:

Strengths are the relatively large sample and the strictly separated training / testing data sets. The results are very good and certainly worth communicating.

R3.1 To get a better sense of the technical data heterogeneity: What are the specifics of the MRI sequences and scanners (vendors, field strength?)

Thank you for pointing this missing information. We added the information about scan manufacturers, field strength, voxel size (reflecting the different DWI protocols) in Table 1. We also added the following paragraph to section 2.1:

“The demographic, lesion and scanner profiles for all the datasets are summarized in Table 1. The distribution of strokes according to arterial territories (MCA > PCA > VEB > ACA) and the demographic characteristics reflect the general population of stroke patients. MRIs were obtained on seven scanners from four different manufacturers, in different magnetic fields (1.5T (~60\%) and 3T), with more than a hundred different protocols. The DWIs had high in plane (axial) resolution (1x1mm, or less), and typical clinical high slice thickness (ranging from 3 to 7 mm). Although a challenge for imaging processing, the technical heterogeneity promotes the potential generalization of the resulting developed tools. “

R3.2 Lesions can not be classified as “stroke”. Stroke is a clinical event. Please use „diffusion abnormality” or „ischemic abnormality” instead of „stroke lesion”

Thank you for this suggestion, we replaced stroke lesion by diffusion abnormality.

R3.3 Discussion (pg. 10/29): “The balanced precision and sensitivity” What does “precision” mean here? Please define.

Precision means $\text{True positives} / (\text{True positives} + \text{False Positives})$. The definitions of all the performance metrics are now included in section 4.8 in the revised draft.

R3.4 Discussion (pg. 11/29): “retrospective radiological evaluation favored the results of the automated segmentation” Can the authors give a quantitative summary of those cases?

Thank you for pointing this missing information. We noticed circumstantial disagreements of our method with the human evaluators in 24 cases (5% of the Testing set). This was added to the 4th paragraph of the Discussion. Retrospectively, the evaluators considered the quality of the machine segmentation better than the human segmentation, particularly regarding to boundary definition, as illustrated in the new Figure 7I.

Reviewer #4

General comments:

In this work, the author proposed a 3D attention-based deep learning model to perform acute ischemic stroke lesion segmentation. The model was trained with a large dataset ($N > 1000$) and the result looks promising. While the proposed idea is interesting and can potentially make a new best benchmark for the acute ischemic stroke lesion segmentation, there are unclear details in the manuscript and there are questions to be addressed. Here are the questions/suggestions:

R4.1. P.1 Abstract: "Our proposed model outperformed generic nets ... ". I understand that authors probably refer "nets" as "neural networks". For audience coming from non-deep-learning field, they may not be aware of this. Please address this unclarity. Several possible suggestions to replace the word are "neural networks", "models", etc.

Thank you for this suggestion. We replaced the "nets" by "networks"

R4.2. P.3 1. Introduction: "The 3D networks suffer from issues ... no prior information about the lesion location". Have authors look into the application of a U-net variation (i.e., 3D) in lesion segmentation tasks?

In the initial phase of this study, we trained/tested UNet variations, like Mnet and UNet 3+, in a subset of 1,000 cases. They did not achieve significant better performance than the original Unet. On the other hand, the models' performance improved with the addition of the attention gates, so we further focused on comparing our attention models with the original UNet. The comparison with other UNet variations is indeed an interesting theme for further evaluation.

For clarification, all the models presented in this study are in 3D.

R4.3. P.4 2.1 Cohort: "From the 2,348 DWIs quality-controlled for clinical analysis," It is impressive that the authors have a large and clean ischemic stroke dataset for training deep learning models. In here, it is unclear that whether 2,348 DWIs are obtained from distinct patients (i.e., one patient may have multiple DWIs), please clarify (From Table 1 I found the answer--- 2,348 patients' DWIs; this is unclear in the text).

Each 2,348 DWIs were obtained from distinct patients. We now clarify this in the text and in the new figure 1, a flow chart of the data included and its analysis. We are glad the reviewer appreciates this work; as predictable, it took several years to complete.

R4.4. P.5 2.1 Cohort: "... and testing dataset ($n=459$), ... testing models.". How did the authors come up with this training ($N=1,390$) and testing ($N=459$) dataset ratio (~33%), which is uncommon to general machine learning model train-test split (e.g., 80-20, 70-30)?

We arbitrarily used 2/3 for training, 1/3 for testing. Because we have large numbers, we could afford a larger testing sample than used in previous studies. The large testing set allowed us to evaluate performance in different subgroups, as shown in Figure 3 and 4 of the revised manuscript.

R4.5. P.6 2.4 Lesion detection and segmentation with deep learning models: “Our proposed 3D network, ... deep supervision.” From figure 1, there are many components in the proposed network. (1) How did the authors come up with the network architecture (e.g., use two consecutive conv-batch-*SeLU* layers for X_{en_i} , etc.); (2) this may be out of this paper scope: have authors investigate other possible gains from the attention-based deep learning models? That is, deep learning model is known to be a black-box approach, which is less appealing to real-life medical application (since we cannot explain why the model performance is good). Recently, there are published research work in ischemic stroke applications (e.g., time-since-stroke prediction, etc.) that attempt to interpret deep learning model features. One advantage of the attention-based model is that the model is trained to automatically identify and extract features from the input images that better predict the outcome. Have authors look into model interpretation methods for the proposed model? This may provide additional information to show that the proposed model is superior than other models (e.g., classical models, DeepMedic, etc.)

We apologize in advance for the long response, but an “historical” perspective of this study will help us to answer these points.

(1) Per previous studies and our own experience with classical methods and 2D networks, we knew an efficient algorithm should be able to 1) locate abnormal voxels, 2) distinguish those from artefacts, 3) refine the lesion boundaries (a challenging task even for human annotators) and, 4) segment lesions of various volumes/profiles with similar efficiency. Focusing on the last goal in particular and inspired by previous studies employing 2D networks with two pathways at different receptive scale levels (like EDD networks (Liang 2017)), we built 3D UNet/MNet models with deep supervision at each level. This aimed to conquer the drawback that although big lesions could be detected at different levels in the original generic UNet, small lesions features could be dropped after down-sampling pathway/max-pooling layer. The inclusion of information from classical methods (here called “IS”) aimed to help the networks to focus on abnormal voxels, even if in small clusters (small lesions). To make the 3D models share information between different scale levels, we used different infra- and inter- connections between convolution layers. This obviously increased the complexity of the networks, raising gradient vanishing issues. To overcome these issues, we introduced batch normalization and some self-normalized activation (*SeLU*). Furthermore, attention techniques were introduced to make the networks focus on the key features of inputs (lesion spatial or morphological features, in this case).

This description is now added to the Methods section of our revised manuscript.

As the reviewer see, we simply combined previous techniques commonly used to solve the challenges we faced (although we are aware that these applications are not deprived of polemic. For example: one might say “Attention is all you need” [Ashish et al. 2017 arXiv:1706.03762] to “Attention is Not All You Need” [Yihe Dong et al. 2021 arXiv:2103.03404]). Arguing, mathematically or empirically, that these are the best choices or that our models’ architecture is the best, is not our aim and is out of our expertise.

Parenthetically, we don’t think network’s architecture design was ever standardized with strong solid mathematical arguments so far. We reinforce that our contribution restricts on offering a reasonably efficient pretrained model, extensively tested, accessible to non-expert researchers, able to support large scale reproducible and objective data analysis.

(2) This is indeed a valuable suggestion. So far, we haven't done any rigorous investigation about the possible gains from the attention-based deep learning models. This is indeed one of our future plans, as it would help us to understand, for example, how to capture lesion's features as humans do (likely based on the four goals mentioned in the beginning of this response). Other future plans include extracting location features according our previously created vascular atlas (NITRC: Arterial Atlas: Tool/Resource Info) and to apply transformer from detecting ischemic lesions to hyperacute ischemic lesions or hemorrhage lesions.

R4.6. Fig 1A. It is unclear about the input data size. I assume the smaller an input rectangle is, the smaller its input size. Please clarify.

The input data size is in (96,112,48,2) voxels or (96,112,48,3) voxels for 2-channel or 3-channel models, respectively. All our models are whole brain 3D models. Sorry for the confusion in the original draft; this information is now specified in the section 4.4, section 4.7.7 and the Figure 2 in the revised manuscript.

R4.7. P. 7 2.5 General comparison of models' performance: "Our proposed model was compared with ..., and DeepMedic." In the introduction, the authors state that "To our best knowledge, Zhang et al. was the first to reveal the potential of the 3D whole-brain dense networks...". Since Zhang et al. model is the first work and the authors' proposed model is also based on 3D information, why did not the authors include the comparison of Zhang et al. model in this work?

Comparing our models to Zhang et al.'s model was indeed our first choice. However, the authors did not provide either their pre-trained model or complete documentation/source codes to reliably reproduced it (as DeepMedic does). Ambiguous information in their paper reduced our confidence on precisely reproducing their pipeline. For example, the image dimensions along the whole networks are unclear, as well as how they combined the patches into the final image for the whole size in the original space (they mentioned "During training, each image was randomly cropped to $24 \times 80 \times 80$ and flipped along the three axes on the fly...."). In addition, the use of DenseNet and some inconsistencies in the description of the images pre-processing and the model validation and testing reduced our enthusiasm about their methods. For example, the description about image "cropping" is obscure and raises questions about the use of the whole brain; they used the statistics calculated from training dataset to normalize testing dataset (so they are not completely independent); the models were "retrained" in the external testing dataset (ISLE), raising questions about generalization and robustness. Finally, as one of our aims is to provide a model readily useful for ischemic lesion segmentation in clinical images, and to compare it with other models that propose to do the same, DeepMedic seem like a proper option.

R4.8. P.7 2.5 General comparison of models' performance & table 2: "This is particularly noticeable in STIR1, ..., particularly in STIR." Are the evaluation metrics (e.g., dice coefficient, etc.) statistically significant? (i.e., the performance of DAGMNET_CH3 model is significantly better than other models in term of dice, etc.). There will be quite a lot of p-values for table 2 when authors address this question; one suggestion is to include several p-values in the main context for major model comparisons and put the rest in the supplementary materials.

Thank you for this suggestion. We included the heatmaps of p-values for the comparison of models' performance (new figure 10). We also include an "*" in Table 2 representing significant differences. In addition, the p-values for major model comparisons were included in the section 2.4 in the new manuscript.

R4.9. Figure 3. Can the authors increase the font size in the figure? Thank you. Also, please clarify the meaning of the first three columns in Figure 3---Why are there examples that the lesion volume is 0, but the dice coefficient is non-zero?

Thank you for this suggestion. We increased the font size in the figures.

The first 3 columns of the original figure 3 (figure 6 in the revised draft) are scatter plots of lesion volumes and DWI and ADC contrasts versus the Dice scores obtained with different models. In other words, they show how the models perform in lesions of diverse volume and contrast. This is now clarified in the figure caption. The lesion volume is never zero, this impression is probably an effect of the large range of values in the y-axis.

R4.10. Table 2. It is exciting to see the proposed model perform generally better than other models. Can authors provide an explanation about why the proposed model (DAGMNet_CH3) has lower spearman correlation of dice and lesion volume than the DeepMedic model? (given the fact that the proposed model outperforms the DeepMedic model in all other spearman correlations)

This is an interesting question, with a somehow contra-intuitive explanation. The Dice metric is, by its definition, biased by the segmented volume. In other words, Dice scores and lesion volumes are naturally positively correlated. If a model performs well in lesions of small volume, as our proposed model does, this correlation is "broken", becoming lower than expected. Therefore, the lower correlation between our model Dice and lesion volume might result from its better performance (significantly higher Dice) than DeepMedic in small lesions and slightly lower Dice (although not significant) in large lesions.

This brief explanation is included in section 2.6 of the revised manuscript

R4.11. 4.6.7 System implementation. "Training our proposed models 200 epochs.... two days." Why 200 epochs were used?

We chose 200 epochs as most experiments of benchmark models converged after 80~120 epochs. Experiments of more complex networks (not included in the results, as they had similar or inferior performance to the models presented and clear overfitting) took even longer (150~180 epochs, some are even more than 3 days to train). Therefore, we used the maximum training epoch as 200, and early terminated it when the validation loss function converged. The model with the maximum validation Dice was saved.

To clarify this and other possible questions about the model's training and validation, we revised the Methods, including more details in subsections 4.7.8. Other specific information can be also found in our response to Reviewer 1, Comment 12.

Reviewers' comments:

Reviewer #1 (Remarks to the Author):

The authors have solved most of my concerns in the revised manuscript. However, it need further minor revision. My minor concerns are list as follows.

(1) In Fig.1, the 2rd-4th grey rectangles are also named as Input, which makes the reader confused. These rectangles denote the features generated by performing down-sampling on the Input (the 1st grey rectangle). Thus, they are not input, only the 1st rectangle is the input.

(2) In general, the IQR is shown in the format: [Q1-Q3] rather than the range. I suggest the Age, NIHSS, lesion volume, lesion contrast and voxel size in Table are shown in the format median[Q1-Q3]. The mean, std, min and max can be removed.

(3) Some symbols ("fn,fwhm") are very strange. a variable is not denoted using two or more alphabets. Use an alphabet with subscript or superscript

(4) in the row: False Positive Subject Detection Rate, why the subject detection rate is 1, however the 95%CI is [0.51,0.59]. it is unreasonable. Please check. The highlights in Table 3 are confused. why no bold metric or two bold metrics in some rows?

(5) How to determine the a1 and a2 in equation (1)?

(6) In the first paragraph, "Strikes of (2,2,4)" seems wrong. stride? Check the writing before again throughout.

(7) In Fig.4, what's the meaning of distribution? Why the maximum probability distribution is >1? I do not think the Fig. 4 can show the proposed method performs well for large lesions. Using the box plots of Dice for small, moderate and large is better.

Reviewer #2 (Remarks to the Author):

I want to thank the authors of the paper for the well-explained responses to the reviewer's comments and deep revision of the manuscript.

The only minor comment that I suggest is regarding page 9, section 2.6, where the authors mention that the Dice score is biased by the lesion volume. I don't think the Dice itself is biased toward lesion volume because even if a lesion volume is small, but there is a good overlap between the predicted segmentation and ground truth, still the Dice is high. However, Dice is sensitive to false positives and this becomes specifically problematic when the lesion is small and only contains a few voxels.

Reviewer #3 (Remarks to the Author):

No further comments

Reviewer #4 (Remarks to the Author):

Thank you for addressing all of my questions.

Reviewer #1:

The authors have solved most of my concerns in the revised manuscript. However, it need further minor revision. My minor concerns are list as follows.

(1) In Fig.1, the 2rd-4th grey rectangles are also named as Input, which makes the reader confused. These rectangles denote the features generated by performing down-sampling on the Input (the 1st grey rectangle). Thus, they are not input, only the 1st rectangle is the input.

Thank you for your suggestions and for reading our paper again. We removed the word "input" from the 2nd-4th rectangles from Fig. 1. We also modified the Figure architecture to make this part clearer.

(2) In general, the IQR is shown in the format: [Q1-Q3] rather than the range. I suggest the Age, NIHSS, lesion volume, lesion contrast and voxel size in Table are shown in the format median[Q1-Q3]. The mean, std, min and max can be removed.

We changed table 1 according to your suggestions, thank you.

(3) Some symbols ("fn,fwhm") are very strange. a variable is not denoted using two or more alphabets. Use an alphabet with subscript or superscript

We changed fn to N_f , and fwhm to W_{fwhm} .

The Figure 1 is also modified accordingly.

(4) in the row: False Positive Subject Detection Rate, why the subject detection rate is 1, however the 95%CI is [0.51,0.59]. it is unreasonable. Please check. The highlights in Table 3 are confused. why no bold metric or two bold metrics in some rows?

One was the median; we agree that this is meaningless for subject-based false positives and removed the median from this row (we apologize we did not notice it before). We also agree that the bolds are distracting and removed the highlighting from Table 1 (for clarification, we had highlighted what we thought were the most crucial results, mentioned and discussed later in the results and discussion sections).

(5) How to determine the a1 and a2 in equation (1)?

They are calculated by fitting the bimodal Gaussian function to the intensity histogram of each DWI.

We clarified by adding the following sentence to the manuscript:

" a_i , b_i , c_i are calculated by least-square fitting the bimodal Gaussian function to the intensity histogram of individual DWI."

(6) In the first paragraph, "Strikes of (2,2,4)" seems wrong. stride? Check the writing before again throughout.

Thank you for catching this mistake. We corrected it and checked the writing throughout again.

(7) In Fig.4, what's the meaning of distribution? Why the maximum probability distribution is >1? I do not think the Fig. 4 can show the proposed method performs well for large lesions. Using the box plots of Dice for small, moderate and large is better.

The distribution is the probability density of Dice scores. So, while the curve's peak can be >1, the area under the curve equals 1. To avoid misunderstanding, we changed the caption and the y-axis labels to "Probability Density"; thank you for pointing the unclearness.

Regarding to the general meaning of Figure 4, while Dice boxplots for small, moderate and large lesions (shown in the right column of Figure 3) show the general tendency of the groups, the distribution shows more details of the performance across Dices. For example, the smoothness of the curves and general absence of "bumps" at low Dices indicate the absence of subgroups of lesions in which the methods fail or perform worse. This information can be lost when the results are summarized in boxplots. Therefore, we don't think Figure 4 is redundant and would like to keep it, with the agreement of the reviewer.

Reviewer #2:

I want to thank the authors of the paper for the well-explained responses to the reviewer's comments and deep revision of the manuscript. The only minor comment that I suggest is regarding page 9, section 2.6, where the authors mention that the Dice score is biased by the lesion volume. I don't think the Dice itself is biased toward lesion volume because even if a lesion volume is small, but there is a good overlap between the predicted segmentation and ground truth, still the Dice is high. However, Dice is sensitive to false positives and this becomes specifically problematic when the lesion is small and only contains a few voxels.

Thank you for your suggestion and for reading our paper again. We changed "Dice is biased" to "Dice is sensitive to false positives".

REVIEWERS' COMMENTS:

Reviewer #1 (Remarks to the Author):

The authors have addressed my concerns in the revised manuscript. No further comments now. this paper can be published.